# Suppression of the Peripheral Immune System Limits the Central Immune Response Following Cuprizone-Feeding: Relevance to Modelling Multiple Sclerosis

**DOI:** 10.3390/cells8111314

**Published:** 2019-10-24

**Authors:** Monokesh K. Sen, Mohammed S. M. Almuslehi, Erika Gyengesi, Simon J. Myers, Peter J. Shortland, David A. Mahns, Jens R. Coorssen

**Affiliations:** 1School of Medicine, Western Sydney University, Locked Bag 1797, Penrith, NSW 2751, Australia; monokesh.sen@westernsydney.edu.au (M.K.S.); m.almuslehi@westernsydney.edu.au (M.S.M.A.); e.gyengesi@westernsydney.edu.au (E.G.); 2Department of Physiology, College of Veterinary Medicine, Diyala University, Diyala, Iraq; 3School of Science and Health, Western Sydney University, Locked Bag 1797, Penrith, NSW 2751, Australia; s.myers@westernsydney.edu.au (S.J.M.); p.shortland@westernsydney.edu.au (P.J.S.); 4Department of Health Sciences, Faculty of Applied Health Sciences, and Department of Biological Sciences, Faculty of Mathematics and Science, Brock University, St. Catharines, Ontario, ON L2S 3A1, Canada

**Keywords:** inside-out, outside-in, oligodendrocytosis, demyelination, gliosis, histology, top-down proteomics, bioinformatics, mitochondria

## Abstract

Cuprizone (CPZ) preferentially affects oligodendrocytes (OLG), resulting in demyelination. To investigate whether central oligodendrocytosis and gliosis triggered an adaptive immune response, the impact of combining a standard (0.2%) or low (0.1%) dose of ingested CPZ with disruption of the blood brain barrier (BBB), using pertussis toxin (PT), was assessed in mice. 0.2% CPZ(±PT) for 5 weeks produced oligodendrocytosis, demyelination and gliosis *plus* marked splenic atrophy (37%) and reduced levels of CD4 (44%) and CD8 (61%). Conversely, 0.1% CPZ(±PT) produced a similar oligodendrocytosis, demyelination and gliosis but a smaller reduction in splenic CD4 (11%) and CD8 (14%) levels and *no* splenic atrophy. Long-term feeding of 0.1% CPZ(±PT) for 12 weeks produced similar reductions in CD4 (27%) and CD8 (43%), as well as splenic atrophy (33%), as seen with 0.2% CPZ(±PT) for 5 weeks. Collectively, these results suggest that 0.1% CPZ for 5 weeks may be a more promising model to study the ‘inside-out’ theory of Multiple Sclerosis (MS). However, neither CD4 nor CD8 were detected in the brain in CPZ±PT groups, indicating that CPZ-mediated suppression of peripheral immune organs is a major impediment to studying the ‘inside-out’ role of the adaptive immune system in this model over long time periods. Notably, CPZ(±PT)-feeding induced changes in the brain proteome related to the suppression of immune function, cellular metabolism, synaptic function and cellular structure/organization, indicating that demyelinating conditions, such as MS, can be initiated in the absence of adaptive immune system involvement.

## 1. Introduction

Currently, there are two competing theories regarding the pathophysiology underlying the initiation of Multiple Sclerosis (MS): ’outside-in‘ and ‘inside-out’ [1,2,3,4]. The former proposes that a dysregulated peripheral immune system leads to an autoimmune response against myelin components of the central nervous system (CNS). The central concept of this theory has been built mainly on the basis of studies using the experimental autoimmune encephalomyelitis (EAE) animal model [5,6,7] and correlation of the end stage of treatment (e.g., paralysis and demyelination) with clinical tests and post-mortem samples from MS patients. In EAE, animals are injected with exogenous antigens such as myelin basic protein (MBP), proteolipid protein (PLP) or myelin oligodendrocyte glycoprotein (MOG) and complete Freund’s adjuvant (CFA), activating peripheral immune cells, including T- and B-cells. When this immune response is combined with breach of the blood brain barrier (BBB) by injection of pertussis toxin (PT), autoreactive adaptive immune cells from the periphery migrate into the CNS leading to degeneration of oligodendrocytes (OLG), demyelination and gliosis [5,6,8,9,10,11]. It is argued that a similar process results in autoimmune cell migration into the CNS of MS patients [7,12,13,14,15]. The EAE animal model is thus the favourite choice of many researchers investigating the autoimmune aspects of MS [16].

There are, however, key differences between the EAE model and clinical MS. First, EAE relies on the use of exogenous antigens (MBP/PLP/MOG), whereas the autoimmune response in humans occurs spontaneously and is only detected following repeated episodes of clinical symptoms [17,18]. Second, the immune reaction in EAE is driven mainly by CD4^+^ T-cells [19,20], whereas in MS, CD8^+^ T-cells predominate [21,22,23]. Moreover, MS is a disease of the human cerebral and cerebellar cortices, whereas the effects of EAE are generally localized to the spinal cord [6,10,24,25,26,27], with largely non-overlapping changes in the brain proteomes being reported in EAE and MS patients [28,29]. Although therapies developed in EAE improve outcomes in animals, these therapies generally have more limited success in clinical MS in terms of halting disease initiation and progression [17,30]. In contrast to this ‘outside-in’ theory, the ‘inside-out’ hypothesis suggests that MS is initiated by an underlying degeneration of OLG and consequent demyelination that leads to the production of endogenous myelin antigens (e.g., peptidyl arginine deiminase, MBP, MOG and PLP) that then trigger an immune response in the CNS [3,4,31]. Histological evidence indicates that the loss of OLG and glial activation can occur in the absence of, or with only a limited number of, peripheral immune cells [32,33,34,35] and myelin injury [36]. The possibility of OLG triggering a secondary adaptive immune response has been reported following long-term diphtheria toxin exposure [37] or following a peripheral immune challenge after short-term CPZ-feeding (termed ‘cuprizone autoimmune encephalitis’ [31]). Cuprizone (CPZ) is synthesised by combining cyclohexanone and oxaldihydrazone [38]. While the mechanism of its toxic actions remain ill-defined, copper chelation [39] and dis-homeostasis of iron, zinc, sodium and manganese have been reported [40,41,42,43,44]. Such ion imbalance leads to endoplasmic reticulum stress, reduced mitochondrial ATP synthesis, and increased production of reactive oxygen and nitrogen species (reviewed in [2]). OLG appear to be preferentially susceptible to CPZ toxicity and likely degenerate due to their high energy demands and lower levels of anti-oxidant enzymes [45,46,47].

Intriguingly, longer-term CPZ-feeding did not evoke a peripheral immune response in the CNS [31,37,48]. Whether the failure of numerous CPZ studies [31,37,48] to observe an immune response associated with long duration feeding (>5 weeks) is due to a toxic effect of CPZ on the peripheral immune system remains unclear [31,37], but CPZ has been shown to have deleterious effects on immune organs like the spleen [49] and thymus [50].

To address this issue, this study compared whether a low (0.1%) or standard (0.2%) dose of CPZ when combined with disruption of the BBB by PT recruited peripheral immune cells into the CNS in mice. At first, a low and a standard dose of CPZ were used for 5 weeks and the amount of oligodendrocytosis (i.e., degeneration or loss of OLG), gliosis and demyelination was quantified; the low dose produced significant demyelination of the corpus callosum (CC), with limited suppression of splenic CD4/8 and no change in overall splenic mass. Having observed that 0.1% CPZ produced comparable oligodendrocytosis, but had less severe effects on the peripheral immune system, in the second study, 0.1% CPZ-feeding was extended to 12 weeks (±PT) to test whether a slower, progressive demyelination (i.e., more reminiscent of MS) and less severe effects on the peripheral immune organs could trigger an adaptive immune response in the CNS. Histological analyses were used to assess oligodendrocytosis, demyelination and gliosis in the CNS, as well as the levels of adaptive immune cells (CD4 and CD8) in brain and spleen.

In a subset of the mice in both experiments, the whole brain proteome was assessed using a well-established ‘top-down’ approach (i.e., two-dimensional gel electrophoresis coupled with liquid chromatography and tandem mass spectrometry) to identify changes in protein abundance correlated with key changes in molecular pathways [49,51,52,53]. To best understand the underlying molecular/cellular processes, a ‘top-down’ proteomic analysis was critical to identifying key protein species or *proteoforms*, the biologically active entities [54,55,56]. Thus, while much is assumed in the literature regarding the actions of CPZ at the molecular level by extrapolation of effects in vitro or at the cellular level, only such quantitative analyses can help to directly understand the underlying effects of CPZ(±PT).

## 2. Materials and Methods

### 2.1. Animals, Feeding, Injection and Monitoring

Seven-week-old male C57Bl/6 mice (*n* = 108) were purchased from the Animal Resources Centre, Murdoch, WA, Australia (www.arc.wa.gov.au) and co-housed (2 mice) in individual ventilated GM500 cages (Tecniplast, Buguggiate, VA, Italy) in the local animal care facility (School of Medicine, Western Sydney University). Animals were allowed to acclimatise for one week to the new environment prior to initiation of CPZ-feeding. Mice were maintained in a controlled environment (12-hour (h) light/dark cycle: 8am–8pm light, 8pm–8am dark, 50–60% humidity and at 21–23 °C, room temperature (RT)) throughout the entire period. Standard rodent powder chow (Gordon’s specialty stockfeeds, Yanderra, NSW, Australia) and water were available *ad libitum.* Oral feeding of CPZ ([Bis(cyclohexanone)oxaldihydrazone, Sigma-Aldrich, St. Louis, MO, USA], 0.1–0.2% *w/w* freshly mixed with rodent chow) was used to induce oligodendrocytosis as previously described [2,57,58,59]. To breach the BBB, the same methods as previously established for EAE were used i.e., 2–3 intraperitoneal (IP) injections of PT [8,60,61,62,63]—but adapted so that the breach of the BBB was timed (i.e., 400 ng on days 14, 16, and 23) to coincide with the reported onset of CPZ-induced oligodendrocytosis, demyelination and gliosis [2,57,58,59]. The efficacy of BBB breach has been shown using immunoglobulin G staining in the CPZ-fed mice [64]. CPZ groups (0.1% and 0.2%) were fed freshly prepared (daily) CPZ in rodent chow for either 5 (n = 10/group) or 12 (n = 12/group) weeks. Age-matched, naïve control (Ctrl, 5-week study n = 10 and 12-week study n = 12) and PT only (5-week study n = 10 and 12-week study n = 12) groups were used. Mice were weighed at the beginning of the studies, weekly throughout, and prior to culling, and the data from both groups (5 and 12 weeks) were combined (Figure 1). Research and animal care procedures were approved by the Western Sydney University Animal Ethics Committee (ethics code: A10394) in accordance with the Australian Code of Practice for the Care and Use of Animals for Scientific Purposes as laid out by the National Health and Medical Research Council of Australia.

### 2.2. Histology and Immunohistochemistry

#### 2.2.1. Tissue Preparation

At the end of each feeding period (5 or 12 weeks), all mice were terminally overdosed with sodium pentabarbitone (250 mg/kg, Lethobarb^TM^, Tory laboratories, Glendenning, NSW, Australia) and perfused with 30 mL of 0.9% saline followed by 50 mL of cold 4% paraformaldehyde (PFA, Sigma-Aldrich) for ~5 minutes (min). Brain and spleen samples were collected and post fixed with 4% PFA at 4 °C for one week and stored in 0.01 M phosphate buffered saline (PBS, Sigma-Aldrich) solution containing 0.02% sodium azide (Amresco, Solon, OH, USA) at 4 °C for ≤1 month or until sectioned. Spleen weights were measured (n = 3 and 5 from 5- and 12-week studies, respectively) and expressed as a function of body weight (Appendix A). Prior to sectioning, whole brain and spleen were immersed in 30% sucrose for 48 h at RT for cryo-protection followed by embedding in 4% gelatine (Chem-Supply, Gillman, SA, Australia) at −20 °C. Brains (5 weeks: 50 µm and 12 weeks: 40 µm) and spleen (20 µm) were sectioned coronally on a Leica cryostat (Leica, Wetzlar, HE, Germany). Sections were transferred to either 6-well plate (Sakura, Torrance, CA, USA) containing cold (5–6 °C) 0.01 M PBS (free floating) or mounted onto 0.5% gelatine-coated slides (Knittel Glass, Braunschweig, NI, Germany) as described previously [65].

#### 2.2.2. Silver Myelin Staining and Analysis

Silver staining of myelin was performed at RT as previously described [66,67]. Briefly, tissue sections were mounted onto 0.5% gelatine-coated slides and air dried for 48 h before immersion in 10% formalin (Sigma-Aldrich) for 2 weeks to increase the contrast of the staining. Sections were stained in a large glass container in parallel to maintain the consistency of staining. Slides were washed with distilled water and pre-treated with lipid-solvent pyridine and acetic anhydride solution (ratio 2:1) for 30 min. Sections were then rehydrated with serial dilutions of ethanol 80, 60, 40 and 20% for 20 seconds (sec) in each step followed by two washes with distilled water. Slides were then immersed in ammonical silver nitrate (Chem-Supply) containing developing solution (0.2% ammonium nitrate, 0.2% silver nitrate and 5% sodium carbonate) for 45 min. Sections were then dehydrated by rinsing sequentially using 20, 40, 60, 80, and 100% ethanol for 20 sec in each step. Sections were cleared by immersing the slides in xylene for 5 min and then sealed using cover slips (Knittel Glass) and ~1 mL mounting medium (Merck, Darmstadt, HE, Germany), and air dried for 72 h. The sections were viewed with an Olympus Carl Zeiss Bright Field Microscope (Zeiss, Jena, TH, Germany) and all images were captured at the same microscope settings (i.e., a fixed exposure time, magnification and illumination intensity). In ImageJ (https://imagej.nih.gov/) software, the region of interest (ROI) was contoured, and the mean optical density quantified (sum of each pixel intensity [range black (0) to white (256)] divided by the number of pixels in the ROI). To quantify the amount of myelin present (which stains black), data were expressed as the reciprocal of the light intensity (i.e., the smaller the value, the lower the myelin content) and normalised to the Ctrl groups (n = 3–5 animals/group and 5–9 sections/animal). The effectiveness of CPZ-feeding is frequently determined by loss of myelin from the midline corpus callosum (MCC), in this study the effectiveness of CPZ-induced demyelination was confirmed in the MCC and lateral corpus callosum (LCC). Anatomical landmarks were identified as described previously [68,69].

#### 2.2.3. Immunofluorescence Staining and Analysis

All staining was performed at RT. Free floating brain (40–50 µm) and slide-mounted spleen (20 µm) coronal tissue sections were washed thrice with warm (40–50 °C) 0.01 M PBS to remove the gelatine and then immersed in 5–10% goat serum (Sigma-Aldrich) for 2 h with agitation at 50 rpm on a shaker table to block non-specific antibody (Ab) binding sites. Sections were then incubated (12 h) with primary monoclonal Ab to either neurite outgrowth inhibitor A (rabbit anti-Nogo A, 1:500, Merck-Millipore, Burlington, MA, USA), glial fibrillary acidic protein (mouse anti-Gfap-Alexa 488, 1:1000, Merck-Millipore), ionized calcium-binding adapter molecule 1 (rabbit anti-Iba 1, 1:1000, Wako, Chuo-Ku, OSA, Japan), anti-cluster of differentiation (rabbit anti-CD4, 1:200, Abcam, Cambridge, UK), or anti-CD8 (mouse anti-CD8, 1:100, Santa-Cruz Biotechnology, Dallas, TX, USA) diluted in 0.01 M PBS containing 0.1% Triton X100 (Tx100). Sections were then washed thrice with 0.01 M PBS and incubated with corresponding Alexa Fluor (either 488 or 555, dilution: 1:500) secondary Ab (diluted in 0.01 M PBS/0.1% Tx100 solution for 2 h while shaking at 50 rpm). Sections were then rinsed thrice with 0.01 M PBS and 1.5 µg/mL Vectasheild™ plus 4′,6-diamidino-2-phenylindole (DAPI, Vector Laboratories, Burlingame, CA, USA) to counterstain nuclei. Slides were sealed with cover slips and stored in the dark at 4 °C until analysis. Images were captured as before (using an Olympus Carl Zeiss Fluorescence Microscope) using the same fixed parameters (exposure and magnification). Quantification was performed using ImageJ, measuring the fluorescence intensity of Gfap, Iba 1, CD4 and CD8 from each ROI as described in silver myelin staining and analysis (*n* = 3–5 animals/group and 5–10 sections/animal). Cells positively stained for Gfap, Iba 1 and Nogo A (and co-stained with DAPI) were counted using the unbiased stereo investigator optical fractionator workflow software [65,70]. To obtain the cell density, total cell number was divided by total measured volume and the data expressed as 10^4^ cells/mm^3^ (*n* = 3–5 animals/group and 5–9 sections/animal).

### 2.3. Two-Dimensional Gel Electrophoresis (2DE) and Analysis

#### 2.3.1. Sample Collection, Homogenisation and Protein Estimation

At the end of each experiment (i.e., 5 or 12 weeks) period, mice (*n* = 5 animals/group) were euthanized by overdose of isoflurane (Cenvet, Blacktown, NSW, Australia) exposure. Whole brains were collected following decapitation and immediately rinsed with ice cold 0.01 M PBS containing a cocktail of protease, kinase, and phosphatase inhibitors (Sigma-Aldrich, [51,52]) to remove any traces of blood. Tissue homogenisation was accomplished by automated frozen disruption using a Mikro-Dismembrator (40 Hz for 1 min, Sartorius, Göttingen, NI, Germany) to facilitate optimal protein extraction [52,71]. Powdered tissue samples were mixed with cold 20 mM HEPES hypotonic lysis buffer (Amresco) containing the cocktail of protease, kinase, and phosphatase inhibitors and vortexed for 90 sec followed by the restoration of isotonicity using the addition of an equivalent volume of ice cold 0.02 M PBS and incubated for 5 min on ice. The samples were then centrifuged (Beckman Coulter, Indianapolis, IN, USA) at 125,000× *g* (using a SW 55 Ti rotor) at 4 °C for 2 h. The resulting first supernatant (SP1) was collected as total cytosolic soluble protein (SP). The pellet was washed with ice cold 0.01 M PBS to extract any remaining cytosolic proteins (SP2) and centrifuged 8 h at 125,000× *g*. The resulting supernatant (SP2) was pooled with the SP1 fraction, and this combined soluble protein fraction was concentrated using an Amicon Ultra-4 centrifugal 3 KD cut-off filter column (Merck-Millipore). To prevent salt interference during the 1^st^ dimension isoelectric focussing step, the resulting SP samples were washed three times using cold 4 M urea (Amresco) buffer supplemented with the inhibitor cocktail. The pellet containing membrane proteins (MP) was resuspended with cold 2DE solubilisation buffer (8 M urea, 2 M thiourea and 4% CHAPS) containing the inhibitor cocktail [71,72]. Total protein concentrations in SP and MP fractions were measured using the EZQ™ Protein Quantitation Kit (Life Technologies, Eugene, OR, USA) according to the manufacturer’s instructions, using bovine serum albumin (Amresco) as the standard.

#### 2.3.2. Protein Separation

At first, proteins were separated based on their isoelectric point (1^st^ dimension) as follows: 100 µg of proteins were loaded onto an immobilised pH gradient (IPG, 7 cm, non-linear, Bio-Rad, Hercules, CA, USA) strips and passively rehydrated for 16 h at RT. Rehydrated IPG strips were placed in the Protean isoelectric focusing (IEF) tray (Bio-Rad) and IEF was carried out in a PROTEAN IEF system (Bio-Rad) for high-throughput protein resolution, initially at 250 V for 15 min which then increased linearly to 4000 V at 50 µA/gel for 2 h, with multiple electrode wick changes during voltage ramping to facilitate desalting. The following parameters were used during IEF: focus temperature: 17 °C, desalting: 15 min, linear gradient: 2 h, holding voltage: 500 V. Following IEF, IPG strips were either stored at −20 °C or immediately resolved in the second dimension using sodium dodecyl sulphate-polyacrylamide gel electrophoresis (SDS-PAGE). Second dimension (2D) was carried out using 1 × 84 × 70 mm 12.5% acrylamide SDS-PAGE gels. Prior to 2D, IPG strips were incubated for 10 min with 130 mM DTT in equilibration buffer (6 M urea, 20% glycerol, 2% SDS and 375 mM tris) followed by 10 min alkylation in equilibration buffer containing 350 mM acrylamide. IPG strips were then inserted on top of the SDS-PAGE gels and covered with warm (40–50 °C) ~300 µL agarose solution (0.5% low melting agarose, Bio-Rad) containing 2% bromophenol blue (Bio-Rad). Electrophoresis was carried out at 4 °C by applying 150 V for 5 min followed by 90 V for 3 h or until the bromophenol dye reached the bottom of the gels as described previously [51,52,71,73].

#### 2.3.3. Protein Fixation and Staining

Upon completion of electrophoresis, gels (5-week study n = 180 gels and 12-week study n = 120 gels) were fixed in 10% methanol and 7% acetic acid solution for 1 h at RT (on a shaker at 50 rpm). Gels were rinsed with distilled water for 3 × 20 min to remove residual methanol and acetic acid followed by staining with 50 mL of high sensitive colloidal Coomassie Brilliant Blue (G-250, Amresco) and continuous shaking, as previously described [52,74,75,76,77,78]. After 20 h, the solution was discarded and stained gels were washed using 50 mL of 0.5 M sodium chloride solution for 3 × 15 min to remove excess Coomassie dye. Scanning of gels was carried out at 100 µm resolution using a Typhoon^TM^ FLA-9000 gel imager (GE Healthcare, Chicago, IL, USA). Excitation/emission wave lengths were 685/>750 nm and the photomultiplier tube was set to 600 V. Gels were preserved in 20% ammonium sulphate (50 mL/gel) and stored at 4 °C until spot excision [52].

#### 2.3.4. Protein Resolution, Detection and Image Analysis

Quantitative analysis of gels was carried out using Delta 2D image analysis software (www.decodon.com/delta2d-version 4.0.8, DECODON, Greifswald, MV, Germany) as described previously [49,52,53,74,78]. Briefly, total spot numbers were calculated from the raw images using the Delta 2D automated spot detection system, while gel edges and the protein ladder were excluded [52,79]. Gel images were warped and fused to generate a master image ensuring consistent spot matching. The fluorescent volumes of individual spots (i.e., protein abundance) were expressed as a function of all spot volumes detected and were measured using Delta 2D to assess changes across the different experimental groups. Four spot inclusion criteria were applied: 1) any changes in spot volumes were detectable in all biological replicates (*n* = 5 animals/group) and their associated technical replicate gels (*n* = 3 gels/fraction/animal, i.e., 15 gels/experimental group); 2) the relative standard deviation for technical replicates did not exceed 30% within individual animals; 3) values had to differ significantly (*p* < 0.05, t-test) between the naïve Ctrl group and at least one test group; and 4) have a fold change of ≥1.5 (increased/decreased fluorescence), to be considered genuine changes, and thus candidates for analysis by LC/MS/MS. These criteria allowed for reliable and reproducible identification of CPZ, CPZ+PT or PT associated proteoform changes. The fold change (*p* < 0.05, one-way ANOVA) of differentially abundant protein spots was calculated by dividing the average grey value (i.e., fluorescence intensity of the protein spot of each experimental group compared to the naïve Ctrl group) and presented in log_2_ scale as fold change relative to Ctrl. Precision Plus Protein Kaleidoscope molecular weight (MW) and 2DE isoelectric point *(pI)* standard (Bio-Rad) calibration gels (*n* = 3) were used to calculate the experimental *pI* and MW of resolved proteoforms. The coefficient of variation (standard deviation/mean) for the MW and *pI* migration was 2.6% and 1.4% for 2DE standards, respectively (*n* = 3), and 4% for MW ladders for experimental gels (*n* = 20). To quantify the gel shift indicative of protein post-translational modification (PTM), experimental MW and *pI* values were plotted relative to theoretical values; significant changes were indicated when the experimental measure fell above or below the 95% confidence intervals of the MW and *pI* calibration curves. The results are plotted as the average for both 5 and 12 weeks combined.

#### 2.3.5. In-Gel Protein Spot Digestion

Protein spots of interest were excised manually and de-stained for 2 × 15 min with 50 mM ammonium bicarbonate (Sigma-Aldrich) solution containing 50% acetonitrile (Sigma-Aldrich). After complete removal of Coomassie dye, gel pieces were dehydrated using 100% acetonitrile. In-gel digestion was carried out by adding 20 µL of freshly prepared trypsin (12.5 ng/µL, Promega Corporation, Madison, WI, USA) to a solution of 50 mM ammonium bicarbonate for 8 h at 4 °C. Digested peptides were removed from the gel by 30 min sonication. The solution was then acidified by the addition of 2 µL of 2% formic acid (Merck-Millipore). The resulting peptide solution was concentrated to 10 µL using a Speed Vac™ vacuum concentrator (1400 rpm for 10–15 min, John Morris Scientific, Chatswood, NSW, Australia) and stored at -80 °C for future use or immediately subjected to LC/MS/MS [52].

#### 2.3.6. Liquid Chromatography Tandem Mass Spectrometry (LC/MS/MS)

The concentrated peptide solutions were analysed by LC/MS/MS using a nanoAcquity ultra performance liquid chromatography system linked to a Xevo QToF mass spectrometer as previously described [49,52,80,81]. In brief, peptide sample solutions (3 µL) were loaded onto a C18 symmetry trapping column (20 mm × 180 µm), and desalted for 3 min at 5 μL/min flow rate using 1% solvent B (LC/MS grade 1% acetonitrile and 0.1% formic acid) in solvent A (Mili-Q water + 0.1% formic acid). The peptides were washed off the trapping column at 400 nL/min onto a C18 BEH analytical column (75 μm × 100 mm), packed with 1.7 μm particles of pore size 130 Å using a 60 min ramped LC protocol. The initial solvent composition was held at 1% B for 1 min followed by linear ramping to 50% B over 30 min. A further linear ramp to 85% B commenced at 31 min and held until 37 min before the solvent composition was returned to 1% B. Separated peptides were analysed using tandem mass spectrometry with a constant cone voltage of 25 V and source temperature of 100 °C, implementing an emitter tip that tapered to 10 µm at 2.3 kV. A data-directed acquisition (DDA) approach was performed, which continuously scanned across the *m/z* range 350–1500 for peptides of charge state 2^+^–4^+^ with an intensity of more than 50 counts/sec, with a maximum of three ions in any given 3 sec scan. Selected peptides were de-isotoped, fragmented and the masses measured. The ramped collision energy profile was set to 15–35 V at low mass and 30–40 V at high mass. The mass of the precursor peptide was then excluded for 30 sec. The DDA was via Masslynx software (version 4.1, Micromass, Manchester, UK) and converted to a peak list file (PKL) format using the ProteinLynx Global Server (Waters, Milford, CT, USA). Data were analysed using MASCOT Daemon (www.matrixscience.com) and queried against the SwissProt and MSPnr100 databases (see www.wehi.edu.au) using delimited and species-specific searches to identify the protein species using the following MASCOT parameter settings: the enzyme trypsin and taxonomy *Mus musculus* (mouse) were fixed. Moreover, no fixed modification was selected whereas variable modifications were carbamidomethyl (C), deamidated (NQ), oxidation (M) and propionamide (C). Only two missed cleavages of lysine or arginine residues were allowed; mass tolerance of parent and MS/MS ions was set to ±0.05 Da and peptide charge state was 2^+^–4^+^. The results of the search were filtered by excluding peptide hits with a *p*-value greater than 0.05. While the SwissProt and MSPnr100 databases were both used to best ensure confirmation of a protein identity, the higher of the two scores was documented in Table 1. When multiple proteins were detected from the same spot, the following criteria were applied to identify the most abundant and thus the most likely to have contributed to the originally detected change in spot volume: 1) The highest MASCOT score (>100) with a sequence coverage ≥5%; and 2) ≥4 unique matched peptides.

#### 2.3.7. Literature Mining and Bioinformatics

A PubMed (www.ncbi.nlm.nih.gov/pubmed/) literature search was carried out for papers published in the English language using the identified canonical protein name with either CPZ, EAE or MS to find literature relevant to molecular/cellular functions. The UniProt (www.uniprot.org) database was used to obtain the gene and UniProt accession number (ID) of the identified protein species and analysis of subcellular localization [82]. A mapping of genes according to their classification and molecular functions was derived from protein analysis through evolutionary relationships (PANTHER, www.pantherdb.org) database using gene IDs of each identified protein [53]. Cellular components, biological processes and physiological pathways of the identified protein species were categorised using the database for annotation, visualization and integrated discovery (DAVID, version 6.8, david.ncifcrf.gov) database. UniProt accession IDs were used in the DAVID database to categorise proteins according to their GO (gene ontology, www.geneontology.org) cellular components, and biological processes. DAVID also characterised the physiological pathways associated with the identified protein species according to the Kyoto encyclopaedia of genes and genomes (KEGG, www.genome.jp/kegg) category [83]. Protein species were further characterised and grouped using the search tool for retrieval of interacting genes/proteins (STRING; version 10, string-db.org) to identify potential protein–protein interactions (PPI, [84]). Using the STRING database, a PPI map was generated in which each node represents a protein and connecting lines represent evidence of association (with line thickness indicating the strength of the potential interaction). Such associations are based on text mining, co-expression, co-occurrence, databases, experiments, neighbourhood and gene fusion of the identified proteins [85,86].

### 2.4. Western Blot (WB)

#### 2.4.1. Sample Preparation

Stored brain, spinal cord and spleen samples (*n* = 3/group) were homogenised in the deep-frozen state as described earlier. Equal ratios (~1 µL/1 µg tissue) of sample and pre-chilled lysis buffer (25 mM Tris, 1 mM EDTA and 1 mM EGTA) containing the inhibitor cocktail were used to solubilize the powdered samples and protein was recovered using centrifugation (at 125,000× *g*, 4 °C, for 1 h). Protein quantification was then carried out as previously described using the EZQ protein quantitation kit (see above).

#### 2.4.2. Procedure

Total protein extract (brain, spinal cord and spleen) and CD4/8 recombinant proteins were resolved by 10% SDS-PAGE (100 V for 2 h at 4 °C) and transferred (100 V for 2 h at 4 °C) onto 0.22 µm pore size polyvinylidene difluoride (PVDF, Merck-Millipore) membrane using transfer buffer containing 25 mM tris, 192 mM glycine and 20% methanol. The membranes were incubated in blocking buffer containing non-fat dry skimmed milk (5% *w/v*, Coles, Hawthorn East, VIC, Australia) and polyvinylpyrrolidone (1% *w/v*, Sigma-Aldrich), in 0.05% Tris buffered saline-Tween 20 (TBST) for 1 h at RT on an orbital shaker (at 50 rpm). Primary Abs for CD4 (rabbit anti-CD4, 1:500, Abcam) and CD8 (mouse anti-CD8, 1:75, Santa-Cruz Biotechnology) were incubated for 1 h at RT. Blots were then washed thrice with 0.05% TBST at 10 min intervals and horseradish peroxidase-conjugated (HRP) secondary Ab (goat anti-rabbit- or goat anti-mouse-HRP: CD8 1:500, Santa-Cruz Biotechnology and CD4 1:2000, Abcam) was added and incubated for 1 h at RT. Chemiluminescent visualization of the transferred proteins was carried out using an enhanced chemiluminescence detection reagent (500 μL/cm^2^ membrane, Merck-Millipore). Blots were scanned for 2 sec on the ImageQuant^TM^ FUJI LAS-4000 biomolecular imager (GE Healthcare). ImageJ software was used to quantify the density of a band of interest on a blot by using a rectangular box to define the band. This band intensity was expressed as a raw value (*n* = 3 bands/animal, *n* = 3 animals/group) and presented relative to Ctrl.

#### 2.4.3. Transfer Efficiency

Replicate 1D SDS-PAGE gels were resolved in parallel and one stained with Coomassie Brilliant Blue prior to, and the other after, transfer onto PVDF membrane to determine the transfer efficiency of proteins. Imaging was carried out using a Typhoon^TM^ FLA-9000 gel imager. The density of the bands (*n* = 3 bands/gel) with the molecular weights corresponding to the known molecular weights of CD4/8 (37 and 50 KD, respectively) were quantified using Multigauge image analysis software-version 3.0 (Fujifilm, Minato-Ku, TYO, Japan).

#### 2.4.4. T-Cell Detection Limits in Peripheral and CNS Tissues

To measure the detection limit of CD4 and CD8 antibody signals in WBs, spleen samples from naïve mice and commercial CD4/8 recombinant proteins (Sino Biological, Wayne, PA, USA) were used. For CD4, the lowest detectable signal was achieved using 5 µg of spleen protein, whereas 10 µg of total spleen protein was needed to detect a CD8 band. The minimal detectable concentrations of the commercial recombinant protein standards were 5 ng and 5 µg for CD4 and CD8, respectively (Figure 3a). These lowest detectable concentrations for both groups (spleen and commercial samples) were used as positive (spike) controls to establish the expected minimal detection of CD4 and CD8 in WBs of total brain protein from the different experimental groups.

### 2.5. Statistical Analysis and Graphing

Statistical analyses were performed using GraphPad Prism-version 7.03 (www.graphpad.com, San Diego, CA, USA) software. Data were analysed using either one or two-way analysis of variance (ANOVA) or an unpaired two-tailed t-test and, where appropriate, Newman-Keuls or Tukey post hoc analyses to determine specific differences among groups. Data are presented as means ± standard error of the mean (SEM), otherwise indicated in the text. Statistical significance was accepted when *p* < 0.05. Figures were assembled using CorelDRAW-version 2018 (www.coreldraw.com, Ottawa, ON, Canada) and Photoshop CS6 (Adobe, San Jose, CA, USA) image processing software. All Nogo A images were adjusted only for colour contrast (Appendix A).

## 3. Results

### 3.1. Body Weight

Mice in all groups gained weight over the duration of feeding, but this was significantly slower in CPZ(±PT)-fed animals (*p* < 0.05, Figure 1). Significant reductions in weight gain started at week 1 after 0.2% CPZ±PT and week 2 after 0.1% CPZ±PT-feeding and continued until week 11. At week 12, 0.1% CPZ±PT groups were significantly (*p* < 0.05) different compared to the Ctrl group but not to the PT group. No direct or combined effect (*p* > 0.05) of PT was found in any group at any time point.

### 3.2. Marked Demyelination, Oligodendrocytosis and Gliosis

In both the 5- and 12-week studies, the Ctrl and PT groups exhibited intense silver staining of myelin in the midline corpus callosum (MCC) and lateral corpus callosum (LCC), whereas the 0.1% and 0.2% CPZ-fed(±PT) groups displayed a marked, concentration-dependent loss of silver staining (*p* < 0.05, Figure 2a; Appendix A). No differences in the extent of demyelination were found between the MCC and LCC regions in any of the groups with either duration of CPZ-feeding [(*p* > 0.05, Figure 2a, (MCC); Appendix A, (LCC)]. Importantly, prolonged 0.1% CPZ (±PT)-feeding for 12 weeks produced a similar amount of demyelination to that seen at 5 weeks with 0.2% CPZ (*p* < 0.05, Figure 2a; Appendix A). Consistent with the significant reduction in silver staining, 5 or 12 weeks of CPZ-feeding produced a significant loss (>90%) of mature, Nogo A positive OLG in the MCC and LCC; feeding with 0.1% was as effective as 0.2% CPZ at inducing OLG loss (*p* < 0.05, Figure 2b; Appendix A). Importantly, there were no differences in OLG loss when using 0.1% or 0.2% CPZ for 5 weeks, and 0.1% CPZ-feeding for 12 weeks produced a comparable loss of OLG to that seen at 5 weeks (*p* > 0.05, Figure 2b; Appendix A). Similarly, there were marked dose-dependent increases (*p* < 0.05) in the number and intensity of Gfap and Iba 1 staining (Figure 2c,d; Appendix A) in the CPZ-fed(±PT) groups compared to Ctrl or PT only animals (which were not different from each other, *p* > 0.05). Taken together, these results indicate that low dose CPZ-feeding for 5 weeks produced an almost complete loss of OLG but a more limited (i.e., slower) demyelination and gliosis response. When this feeding regime was prolonged for 12 weeks, it produced comparable changes to those seen at 5 weeks using 0.2% CPZ. PT had no direct or synergistic effects on any of the histological parameters studied at either time point.

### 3.3. Detection and Localisation of CD4 and CD8 T Cells

Immunofluorescence staining of brain sections failed to detect CD4^+^ and CD8^+^ positive cells in the CC in CPZ(±PT) groups (Appendix A, even when using high antibody titres and long incubation times). This was not due to the lack of antibody sensitivity, as immunofluorescent CD4 and CD8 positive cells were seen in histological sections of spleen (Appendix A). Likewise, CD4/8 signals in whole brain protein samples were undetectable by western blot (WB) analysis (Figure 3c), even at high protein loads (up to 120 µg). This was not due to the lack of protein transfer from SDS-PAGE gel to PVDF membrane, as transfer efficacy at the molecular weights corresponding to CD4/8 (37 and 50 KD, respectively) were 93 ± 1.6% and 98 ± 0.7%, respectively (Appendix A). Furthermore, CD4/8 signals were also detected by WB analysis when spleen (Figure 3a,c) and EAE spinal cord were used as positive controls (Appendix A; EAE was induced using an established method [8]). The capacity and sensitivity of WB to detect CD4/8 signal was also confirmed by other control experiments in which brain homogenates were spiked with spleen homogenate or commercially available CD4/8 recombinant proteins (Figure 3b). Notably, measurement of splenic weight showed a significant (*p* < 0.05) decrease in splenic mass (37 ± 0.1%) in 0.2% CPZ-fed(±PT) groups whereas no change was observed in 0.1% CPZ(±PT) groups at 5 weeks (Appendix A). In contrast, a significant (*p* < 0.05) reduction of splenic mass was observed in 0.1% CPZ(±PT) groups at 12 weeks (Appendix A). Moreover, 5 weeks of CPZ-feeding also resulted in a significant dose-dependent reduction in CD4/8 in spleen compared to Ctrl (Figure 3d, 0.2% > 0.1%, *p* < 0.05). Following 5 weeks of 0.1% CPZ-feeding(±PT), the reduction of CD4 (11 ± 0.03%) and CD8 (14 ± 0.03%) signal intensity was less marked than that seen with 0.2% CPZ(±PT) groups (CD4 44 ± 0.03% and CD8 61 ± 0.04%). Following 12 weeks of 0.1% CPZ(±PT), further reductions in spleen CD4 (27 ± 0.01%) and CD8 (43 ± 0.02%) signal intensity and splenic atrophy (33 ± 0.1%) were observed. In addition, immunofluorescence staining of spleen sections (n = 10 sections/animal and n = 3 animals/group) indicated a significant (*p* < 0.05) reduction of CD4 and CD8 in the 0.2% CPZ-fed group (Appendix A).

### 3.4. Brain Proteome Changes

All samples yielded well-resolved proteomes encompassing the full MW/*pI* range of the gels. Representative images of soluble (SP) and membrane (MP) proteomes from Ctrl and 0.2% CPZ-fed mice are shown in Figure 4a. A total of ~1650 consensus spots (i.e., protein spots that resolved consistently and were analysed across all gels) were detected from the combined analyses of whole brain soluble and membrane proteomes (Figure 4a; Appendix A). The spot quantification from the different groups from both time points is summarized in Appendix A. The different groups yielded comparable numbers of resolved protein spots at 5 weeks (*p* > 0.05), and this was also true for the 12-week samples (*p* > 0.05, Appendix A). In total, 845 ± 8 and 793 ± 11 spots were resolved from the soluble and membrane fractions, respectively, in the 5-week study, whereas 824 ± 11 and 717 ± 3 spots were resolved from the soluble and membrane fractions, respectively, in the 12-week study. Database hits of high-quality and confidence were returned following LC/MS/MS analysis and identified 33 different proteoforms from these spots including 73% in the membrane proteome and 27% in the soluble proteome (Table 1).

Table 1 summarizes the best identified proteoforms within each spot that displayed a 100% reproducible change across technical (*n* = 3 gels/fraction) and biological (*n* = 5 animals/group) replicates. All identified proteins had a MASCOT score exceeding 100, with 46% between 100–200; 21% between 201–300; 9% between 301–400; 6% between 401–500; and 18% exceeding 500. Moreover, each identification was based on at least 4 unique peptides: 61% based on 4–10 peptides; 27% based on 11–20 peptides; and 9% based on over 30 peptides. Similarly, sequence coverage was always ≥5% with 5–10% for 6 proteins; 11–20% for 11 proteins; 21–30% for 8 proteins; 31–40% for 5 proteins; and >50% for 3 proteins (Table 1). The combination of high MASCOT scores and the presence of ≥4 unique peptides with high coverage highlight the high confidence of the protein identifications.

As also shown in Table 1 and Figure 4b, several of the identified proteoforms displayed a mismatch between their theoretical and experimental MW and *pI,* indicative of post-translational modifications (e.g., phosphorylation and glycosylation). Only, 1 proteoform showed an increase whereas 19 (58%) showed a decrease and 13 (39%) showed the same experimental *pI* relative to the theoretical *pI*. In contrast, twenty-five (75%) proteoforms increased, 4 (12%) decreased, while another 4 remained unchanged in their experimental MW relative to theoretical MW. Interestingly, a subset of proteoforms was found showing an approximate doubling of the experimental MW relative to theoretical—hexokinase 1, aconitate hydratase, ATP synthase subunit-α, ogdhl protein, tyrosine-tRNA ligase and dynamin 1—potentially indicative of dimerization, whereas an approximate tripling of MW may indicate calreticulin trimers (Table 1) suggesting a possible increase in oligomerization/self-association of proteoforms due to CPZ-feeding(±PT).

Statistical comparisons of the spot intensities between all groups revealed significant and reproducible changes (*p* < 0.05) in 33 spots across Ctrls and experimental groups in the 5- and 12-week studies (Figure 5). Among the identified proteoforms, not all changes were sustained, or shared, between the 5- and 12-week studies (Appendix A). In total, 23 shared proteoform changes were found in both the 5- and 12- week studies (i.e., ≥1.5-fold change in both studies), whereas 7 proteoforms that changed in the 5-week study (≥1.5-fold) did not in the 12-week study (<1.5 fold) and 3 other proteoforms were changed by 12 weeks (≥1.5-fold) but not at 5 weeks (<1.5 fold). The presence of changes at 5 weeks that were not evident at 12 weeks indicates that, relative to their age matched controls, such changes are time-dependent and resolved by 12 weeks. Whether this return to control levels during prolonged feeding with 0.1% CPZ is due to compensatory mechanisms, aging or other processes remains unknown.

**Key**: MP, membrane protein; SP, soluble protein; MW, molecular weight; *pI*, isoelectric point; S, Swiss-Prot; M, MSPnr100; PT, pertussis toxin; 0.1, 0.1% CPZ; 0.1PT, 0.1% CPZ+PT; 0.2, 0.2% CPZ; 0.2PT, 0.2% CPZ+PT; W, week; ×, unchanged; ↑, increase; ↓, decrease; -, not found or investigated (details are shown in Figure 5). Some of the spots contained more than one clearly identifiable protein; presented here are the hits with the highest score, coverage and peptide count. UniProt and gene IDs were derived from the UniProt database. MASCOT score, sequence coverage, theoretical (MW/*pI*), and unique peptides number were acquired from the MASCOT database search. Experimental (MW/*pI*) was derived from 2D gels of identified protein spot. References are from the published literature in PubMed on CPZ, EAE and MS and used to compare currently identified proteins with the existing literature.

### 3.5. Literature Mining

Literature mining via PubMed was used to assess the likely function(s) of identified proteoforms. This confirmed 8 proteoforms (creatine kinase U-type, glutamate dehydrogenase 1, vesicle-fusing ATPase, propionyl co-enzyme A carboxylase-β, tyrosine-tRNA ligase, actin-related protein 2/3 complex subunit 5, charged multi-vesicular body protein and adipocyte plasma membrane-associated protein) not previously related to CPZ, EAE or MS, and 25 proteins (but not specific proteoforms) previously associated with CPZ (8), EAE (13) and MS (16) studies (Table 1).

### 3.6. Biological Processes and Pathways

The 33 differentially expressed proteoforms were subjected to bioinformatic analysis using PANTHER, DAVID, UniProt and STRING for association with protein classes, molecular functions, physiological pathways, biological processes, cellular components, subcellular localizations and protein–protein interactions. Analysis using PANTHER indicated that the main protein classes were enzyme modulator (15%), nucleic acid binding (10%), oxidoreductase (10%) with 19% unclassified (Appendix A). Molecular function analysis using PANTHER indicated catalytic activity (40%) and binding (26%) roles with 18% unclassified (Figure 6a). However, the potential functions of all the proteoforms were inferred by literature mining for information on the canonical proteins (see *Discussion*). Physiological pathway analysis associated 26% of the proteins with KEGG metabolic pathways categories using DAVID (Figure 6b). GO biological process analysis showed the main categories were oxidation-reduction (13%) and transportation (13%), with 15% unclassified (Appendix A). Moreover, GO cellular component analysis (Figure 6c) indicated that most of the proteins were either cytoplasmic (25%), extracellular exosome (21%), mitochondrial (20%) or related to the myelin sheath (20%). Furthermore, subcellular localization analysis using UniProt revealed the main categories to be mitochondrial (31%) and cytoplasmic (26%; Appendix A). The identified proteoforms were also analysed using STRING software, providing PPI maps; this indicated that 15 of the 33 proteoforms (hexokinase 1, fructose-bisphosphate aldolase C, aconitate hydratase, succinate dehydrogenase flavoprotein subunit, malate dehydrogenase, NADH dehydrogenase iron-sulfur protein 2, ATP synthase subunit-α, voltage-dependent anion-selective channel protein, aspartate aminotransferase, ogdhl protein, isovaleryl-CoA dehydrogenase, tyrosine-tRNA ligase, glutamate dehydrogenase 1, propionyl co-enzyme A carboxylase-β and creatine kinase U-type) were potentially involved in major ‘functional interactions’ particularly with regard to the metabolic proteins (Figure 6d). Strong one-to-one connections were also indicated between synaptic (rab GDP dissociation inhibitor-α, rab GDP dissociation inhibitor-β, vesicle-fusing ATPase, calcium/calmodulin-dependent protein kinase type II subunit-α and dynamin 1) and structural (glial fibrillary acidic protein and neurofilament light polypeptide) proteoforms. This therefore, also revealed a second cluster of structural and signalling proteoforms. Among the connected proteoforms, 38% showed the highest connecting value (0.9), and 29% and 33% were high (0.7) and medium (0.4), respectively; no low value (0.15) connections were found.

Although the addition of PT injections during CPZ-feeding did not enhance the extent of OLG degeneration, demyelination and gliosis, it did produce several proteoform changes (Figure 5 and Figure 7; Appendix A). This is the first study to assess changes in the mouse brain proteome profile when the BBB is compromised by PT. Following 5 weeks of CPZ-feeding, PT injection more than doubled the number of significant CPZ-associated changes (18–46%), with PT alone contributing 15% of all proteoform changes, including fructose-bisphosphate aldolase C, NADH dehydrogenase iron-sulfur protein 2, ATP synthase subunit-α, rab GDP dissociation inhibitor-β, septin-2 and 5, guanine nucleotide-binding protein G(o) subunit-α, adipocyte plasma membrane-associated protein and leukocyte elastase inhibitor A. Likewise, in the 12-week study, the small number of proteins identified following CPZ-feeding (6%) increased following the addition of PT (52%), with PT alone contributing 21% including aconitate hydratase, succinate dehydrogenase flavoprotein subunit, NADH dehydrogenase iron-sulphur protein 2, isovaleryl-CoA dehydrogenase, tyrosine-tRNA ligase, syntaxin-binding protein 1, glial fibrillary acidic protein, neurofilament light polypeptide, septin-2 and calreticulin. Statistical comparisons of the CPZ(+PT) groups with the PT only group (Appendix A) revealed that at least 50% of the PT-mediated proteoform changes were either increased (or decreased) when combined with CPZ-feeding; in the 5-week study, 25% increased (27% decreased) whereas, in the 12-week study 52% increased (6% decreased).

## 4. Discussion

The present study was designed to test whether CPZ-induced oligodendrocytosis, when combined with PT-induced BBB disruption, could induce an ‘inside-out’ activation of the immune system causing infiltration and detection of CD4/8 immune cells into the brain parenchyma. The effectiveness of CPZ-feeding was confirmed by the reduced weight gain in these groups, the almost complete demyelination of the corpus callosum using the standard feeding paradigm of 0.2% CPZ for 5 weeks, and changes to the brain proteome [58,59,108,109]. The results also demonstrated that 0.2% or 0.1% CPZ(±PT) produced comparable oligodendrocytosis but dose- and time-dependent demyelination and gliosis within the corpus callosum, indicating that 0.1% CPZ is as effective as higher doses when fed for a longer period of time. The presence of comparable oligodendrocytosis, but less marked demyelination and gliosis following 5 weeks of 0.1% CPZ-feeding, suggested a slower transition between oligodendrocytosis and demyelination and/or better clearance of myelin debris. In this transition state, limited gliosis may impede subsequent remyelination [110] and induce a slow, progressive demyelination reminiscent of MS. CPZ-induced a dose dependent (0.2% > 0.1%) atrophy of the spleen and extending the CPZ-feeding resulted in time dependent atrophy in 0.1% CPZ(±PT)-fed groups as well. Likewise, CPZ produced a dose-dependent (0.2% > 0.1%) suppression of CD4/8 in the spleen. Prolonged low-dose feeding of CPZ for 12 weeks did not cause a further decrease in CD4 compared to 5 weeks; however, with prolonged feeding, the reduction of CD8 deteriorated to levels seen at 5 weeks using the standard CPZ-feeding. Our results are thus consistent with a previous study where apoptosis of CD4 and CD8 positive cells and atrophy of thymus were observed following 0.2% CPZ-feeding [50]. In contrast, 5 weeks of 0.1% CPZ-feeding produced less CD4 and CD8 suppression and no splenic atrophy, suggesting this lower dose may be a better model to examine the aetiology of MS.

Following CPZ-feeding(±PT) in either experiment, there were no detectable CD4 or CD8 signals in brain tissue. There are several possible reasons for this: firstly, insensitivity of the immunohistological or biochemical assays. This is unlikely, as WB sensitivity was confirmed by spiking whole brain homogenates with biological (spleen) and commercially available CD4/8 recombinant proteins, and both high protein loads (60–120 µg) and high antibody titres were used [111]. Additionally, these results are consistent with previous studies using immunofluorescence and flow cytometry [48,112] as well as BBB disruption using ethidium bromide or lysolecithin [48]. Secondly, the BBB was breached during weeks 1–2 of CPZ-feeding, and this is a transient event [113]. However, whether infiltration occurred at this early time point, but resolved before the 5- or 12-week endpoints remains unknown. Was the lack of a detectable adaptive immune response the result of failure to generate an antigen? This seems unlikely, as marked gliosis, demyelination and oligodendrocytosis occurred. Alternatively, was there a direct effect of CPZ on peripheral immune function? This would be consistent with the observed suppression of CD4/8 signal in the spleens of CPZ-fed animals. It was recently reported that combining two weeks of 0.2% CPZ-feeding with an ‘immune booster’ (CFA) produced a secondary CD3^+^ (pan T-cell marker) response, a response that was not observed with 0.2% CPZ-feeding alone or when CPZ-feeding was continued [31]. Consequently, it can be expected that prolonged CPZ-feeding is unlikely to facilitate infiltration of immune cells into the CNS as continued feeding would result in sustained immune suppression (and reduction of mitochondrial ATP production) which is known to lead to declining health and death of mice. Likewise, extending observations beyond the cessation of CPZ-feeding is unlikely to reveal immune cell infiltration since cessation of CPZ-feeding results in spontaneous remyelination, reduction in myelin debris (a potential antigen), and cessation of glial activation (i.e., the antigen presenting cells) (reviewed in [2,39,57,58,114]).

Notably, the prolonged CPZ-feeding preferentially suppressed CD8 signal intensity in the spleen, and these are the cells that predominate in human MS CNS pathology [22]. These observations are further supported by the large number of changes in spleen (n = 22) and peripheral blood mononuclear cell-derived (n = 5) proteoforms in the CPZ-fed animals [49]. In the spleen, the vast majority (87%) of these changes included membrane associated structural and metabolic proteins suggesting perturbation of adaptive immune function [49]. The spleen and thymus are responsible for the maturation, selection and proliferation of T-cells [115,116], key functions that are impeded by the ion dishomeostasis induced by CPZ [41,42,43,44]. Such effects on the peripheral immune system may explain why the severity of disease and extent of peripheral immune involvement in EAE and Theiler’s murine encephalomyelitis were suppressed by CPZ [117,118,119]. Although, there is no evidence as to whether T-cells become functionally inactivated (’T-cell anergy’; [120,121]) due to the suppression of adaptive immune organs (e.g., thymus) following CPZ-feeding, there is some evidence indicating that CPZ may otherwise affect T-cell functions. Copper plays both direct and indirect roles (via interleukin-2) in the maturation and production of functional T-cells [122]. Consequently, it can be argued that CPZ, like other copper chelators (e.g., 2,3,2-tetraamine), can reduce T-cell function [123,124]. Moreover, it has been shown that when lymph node cells harvested from CPZ-fed mice are exposed to neuroantigens (e.g., concanavalin A and MBP) in vitro, there is reduced cellular proliferation. Likewise, reduced T-helper cell 1 cytokines (e.g., interferon-γ and tumor necrosis factor-α) and CD4 T-cell response to interleukin-17 are observed in EAE mice fed with CPZ [119]. Together, these findings indicate that CPZ can modify the functional capacity of T-cells, and this may potentially affect migration into the CNS. 

The suppression of CD4/8 in the spleen indicated that CPZ has direct effects on the peripheral immune system that may limit the maturation and migration of adaptive immune cells to the CNS. Moreover, calcium/calmodulin-dependent protein kinase type II subunit-α, a protein known to play a role in CD8 T-cell proliferation and the transition to a cytotoxic phenotype [125,126] decreased in abundance following CPZ-feeding. Likewise, leukocyte elastase inhibitor A, a protein that supresses proteases including those released by T-cells [127], also decreased in abundance. These findings extend earlier proteomic analyses [49] that found CPZ reduced the abundance of protein disulphide isomerase (subunits A2, A3 and A6) in spleen, a protein involved with folding and assembly of functional major histocompatibility complex class I molecules [128]. In addition, the CPZ-induced dysregulation of mitochondria identified in the present study presumably extends to the immune system as several studies have shown that mitochondrial dysfunction leads to the suppression of T-cell function and a compromised immune system [129,130]. Collectively, these findings indicate that CPZ has suppressive effects on the capacity of the peripheral immune system to launch a response, making it more difficult to address the ‘inside-out hypothesis’. Therefore, strategies to overcome peripheral immune organ suppression should be the next stage of investigation in evaluating the ‘inside-out’ hypothesis of MS using the CPZ model.

While there is growing evidence that MS may initiate as a slow, low-grade primary oligodendrocytosis, a substantial gap still exists in our understanding of the molecular mechanisms underlying the fundamental susceptibility to and initiation of oligodendrocytosis [1,2,3,4]. Therefore, a quantitative ‘top-down’ proteomic approach was carried out to resolve protein species from whole brains of the CPZ-fed(±PT) mice. To date, proteome analyses of MS have mainly assessed tissue from late-stage post mortem brain samples and cerebrospinal fluid [89,96] or EAE animals [88,93,100,105]; these analyses provide useful insight into the ‘final readout’ of the disease or its autoimmune aspects [131,132,133] but not necessarily insight into the processes involved in disease aetiology and progression, including the role of oligodendrocytosis [1,49].

Here, CPZ-mediated oligodendrocytosis and glial activation were confirmed using histology, and the changes in protein abundance were quantified using a high sensitivity ‘top-down’ analysis rather than the more common ‘bottom-up’ (i.e., ‘shotgun’) analyses; this is critical, as the ‘top-down’ approach has the highest inherent capacity to resolve intact proteoforms (i.e., isoforms, splice variants, and post-translationally modified species) as well as provide better sequence coverage, and assessment of lower molecular weight species, with a high degree of consistency across technical and biological replicates [54,55,134]. This approach thus detected changes in 33 proteoforms (16 metabolic, 7 synaptic, 5 structural and 5 signalling), of which 8 have not been previously associated with CPZ, EAE or MS. Furthermore, it also provided high quality confirmation of changes in the abundance of 25 proteoforms of proteins previously related to MS or animal models of the disorder. In contrast to earlier work [49,91,97,99,135], this study used three technical replicates for each of the 5 biological replicates per experimental group, ensuring that only the most reproducible changes in proteoform abundance were assessed. Consequently, in contrast to the detection of ~700–1200 spots in previous studies [49,88,100,105], ~1650 protein spots per condition were resolved in this study.

The most striking observation to emerge from the proteomic analysis was the large number of changes in metabolic proteoforms involved in glycolysis, Krebs cycle and oxidative phosphorylation pathways and the regulation of mitochondrial function. Bioinformatics platforms (DAVID, PANTHER, UniProt and STRING) further revealed the likely association between metabolic dysregulation and CPZ-feeding. Understanding the molecular pathways and potential PPI is important to understanding how dysregulation of biological processes may lead to a disease [132,136]. It is thus notable that over 80% of proteins do not function alone but in complexes [136]. The data here suggested that 79% of the identified proteoforms may be clustered in two complexes, with 58% being metabolic in origin, which is consistent with previous observations [136,137]. The strong PPI highlight the likely cross-talk and shared biochemical reactions among the metabolic proteoforms [136]. Interestingly, protein-to-protein connection analysis revealed that malate dehydrogenase may have 11 connections with other identified proteoforms, which was the highest seen, and that the ATP synthase subunit-α had the second highest with 7 connections (Figure 6d). The central placement and increased inter-connectedness of malate dehydrogenase or ATP synthase subunit-α with other proteoforms implies that these proteoforms may be pivotal for CPZ-induced metabolic dysregulation. This explanation is further supported by the increased abundance of ATP synthase subunit-α and malate dehydrogenase (in contrast to the decreased abundance of most of the other identified metabolic proteoforms), perhaps as a compensatory response to the CPZ-induced metabolic perturbations. The increased abundance of ATP synthase subunit-α after CPZ-feeding leads to the accumulation of protons in the inter-membrane space of mitochondria and increased generation of reactive oxygen species [91]. Likewise, an increase in the abundance of aconitate hydratase (6 PPI connections, Figure 6d), an iron-sulphur protein containing 4Fe-4S clusters, is likely to regulate iron-induced oxidative stress in the CPZ model. Disruption of iron metabolism is linked to increased oxidative stress and lipid peroxidation [138]; CPZ-feeding is associated with the dysregulation of iron [43] and thus increased oxidative stress results in accentuation of mitochondrial perturbations [139]. Moreover, we observed a synergy between CPZ-feeding(±PT) with regard to synaptic protein dynamin 1 (increased at 5 weeks but decreased at 12 weeks) suggesting that CPZ-feeding interferes with the fission and fusion dynamics of mitochondria. The initial increase in dynamin 1 is consistent with mitochondrial response to metabolic stress, wherein the mitochondria divide, generating new functional mitochondria, and target damaged mitochondria for autophagy [140]. Perturbations to the fission and fusion dynamics of mitochondria are consistent with the formation of mega-mitochondria reported in the CNS, in particular in OLG [141,142]. The current data are consistent with previous investigations [49,50,91,99] corroborating the hypothesised link between mitochondrial dysregulation with CPZ-feeding and the emergence of structural and functional abnormalities that may predispose OLG to degeneration and death [1].

This susceptibility of OLG may be explained, in part, by the intense energy requirements associated with the production and maintenance of the expansive myelin sheath [143,144]. The high metabolic rate of OLG means that the increased production of reactive oxygen (and nitrogen) species, coupled with their low levels of anti-oxidants (e.g., glutathione [145], metallothionein [146] and manganese superoxide dismutase [147]) predisposes them to oxidative injury [45,148]. In addition, increased abundance (in 5- and 12-week studies) of the endoplasmic reticulum (ER) stress-related chaperone protein calreticulin may be associated with unfolded protein responses that enhance oligodendrocytosis [144,149]. This finding extends previously observed changes in ER proteins such as ribosome-binding protein 1, endoplasmin [49] and heat shock protein [99] reinforcing the role of ER stress in the CPZ-fed mice. The presence of heat shock protein and activating transcription factor 4 in MS [89,150,151] and EAE [105,150] indicate the association of protein misfolding and ER stress with OLG degeneration and demyelination.

The apparent oligomerization of some proteoforms in response to CPZ-feeding(±PT) may be indicative of toxic protein aggregation [152,153], effects that have also been observed in an EAE study [154] and MS patients [155]. The oligomerization of proteoforms has been documented in other studies, as with calreticulin and dynamin, assessed using SDS-PAGE or crystallographic analysis [156,157,158]. Likewise, the divergence of molecular weight was also observed in other proteomic studies of CPZ-fed mice (protein phosphatase 1G, 59.38 vs. 104.7 KD) and MS cerebrospinal fluid (albumin, 67 vs. 180 KD and alpha 1 antitrypsin, 47 vs. 100 KD) [49,159]. Indeed, changes in theoretical vs. experimentally observed isoelectric point have also been reported in previous studies on CPZ (ornithine carbamoyltransferase, 8.81 vs. 6.9 and ribosome-binding protein 1, 9.35 vs. 5.0) [49], EAE (septin-8, 5.7 vs. 6.4 and cytochrome c oxidase, 9.2 vs. 5.8) [100] and MS (Ig kappa chain NIG 93 precursor, 8.1 vs. 6.5 and albumin, 5.5 vs. 9) [159,160]. Naturally, none of this key data, upon which to better design future studies to identify critical proteoforms (rather than just amino acid sequences), would be routinely available using any other current approach.

Within the demyelinated regions, the hypertrophy of astrocytes and microglia, compounded by increased microglia numbers, may intensify the local energy imbalance and further compromise the function of OLG. Moreover, hypertrophied microglia and astrocytes may diminish the supportive roles played by glia at synapses leading to a pre-disposition to excitotoxicity [161,162,163]—a disturbance further implicated by changes in other synaptic proteoforms identified in this study.

The abundance of synaptic-regulatory proteoforms was either decreased (rab GDP dissociation inhibitor-α/β and vesicle-fusing ATPase), elevated (charged multivesicular body protein) or displayed opposing changes (syntaxin-binding protein 1) across the two time points—changes in synaptic function that may contribute to alteration of mood or behaviour in humans or animal models [164,165,166]. Likewise, the changes of abundance of proteins such as calcineurin, calbindin 2 and parvalbumin-α, involved in neurotransmitter release, has also been reported in EAE [100]. Although CPZ-feeding induce a wide range of behavioural deficits including motor, anxiety and cognition [reviewed by 2]; the present proteome analysis did not seek correlation with behavioural phenotypes, other studies have argued that CPZ-induced alteration of proteoforms involved in neurotransmitter release can result in cognitive decline [166] or increased climbing, rearing and ambulatory behaviours [99,108,109]. This study also revealed a marked change in neuronal and glial structural proteoforms, including Gfap, neurofilament light polypeptide, actin-related protein 2/3 complex subunit 5, and septins-2 and -5, at both time points, indicating axonal and glial remodelling. Consistent with the capacity for CPZ to induce the hypertrophy of astrocytes and increase microglia in the innate immune system (Figure 2a; Appendix A), a marked increase in proteoforms involved in structural (Gfap), signalling, and inflammatory pathways were observed.

For some proteoforms, breaching the BBB negated the effects of 5 weeks of CPZ-feeding (e.g., septin-5, creatine kinase U-type and 14-3-3 protein epsilon). The effects of PT alone, or in combination with CPZ, indicate that PT did have an effect on the brain proteome, likely, in part at least, by altering the capacity of the BBB to regulate access to the CNS [60,113,167]. This is the first study to document the effects of giving PT alone on the CD4/8^+^ cell migration and the whole brain proteome. In EAE, when PT is given together with adjuvant (CFA) and antigen stimulation (e.g., MOG), increased disruption of tight junctions at sites of perivascular inflammation and demyelination occurs [63]. In addition, increased rates of relapsing-remitting episodes [60], increased infiltration of serum albumin in the spinal cord [131] and suppression of the anti-inflammatory interleukin-10 [168] are observed. However, the extent to which these changes are attributable to PT alone, or the combined treatments used to induce EAE, remains un-documented. However, other studies have shown that PT alone evokes changes in BBB function leading to increased protein infiltration into the brain (~15 days; [169]), or disruption of G-protein function (~40 days; [170]) following PT administration. In the present study, repeated injections of PT during the second and third weeks of CPZ-feeding resulted in proteoform changes after 5 or 12 weeks indicating that PT injections alone have long-term effects at least on the brain proteome. Moreover, the proteomic analysis highlighted proteoform differences between the CPZ vs. EAE models; specifically, CPZ-feeding resulted in increased guanine nucleotide-binding protein G(o) subunit-α, glutamate dehydrogenase 1 and malate dehydrogenase, which decrease in EAE [88,94,105], perhaps reflecting the different underlying aetiologies (potentially including changes in specific proteoforms). Whether these opposing changes result specifically from the use of peripherally administered exogenous myelin antigens (e.g., MBP, PLP and MOG) in EAE or endogenously generated antigens (i.e., myelin debris) in the brain of CPZ-fed mice remains unclear but would seem likely.

Despite our rigorous efforts to minimize the experimental variables, we acknowledge certain inherent limitations in the analytical approach. This study relied on only two time points (i.e., 5 and 12 weeks) of CPZ-feeding(±PT), which did not allow us to determine if or when the identified proteoforms returned to baseline nor to correlate proteome changes with the initiation of oligodendrocytosis, demyelination and gliosis. Moreover, the sub-femtomole in-gel detection sensitivity may well have missed significant changes in very low abundance species, although these remain a substantial issue with all available analytical approaches if high quality final identifications are a serious expectation [78]. Furthermore, reliance on existing databases that address only amino acid sequences, as well as an apparent developing reliance in the field for online bioinformatics platforms that also largely address only what is known about canonical proteins [171,172,173], tends to further emphasize the fundamental importance of developing even more sensitive analytical approaches to routinely quantifying and fully characterizing proteoforms in order to provide the most direct understanding of the molecular mechanism underlying human diseases like MS.

## 5. Conclusions

This study confirmed that CPZ-feeding(±PT) in mice induced dose- and time-dependent oligodendrocytosis, demyelination and gliosis, but was not associated with any detectable invasion of peripheral adaptive (CD4/8) immune cells into the CNS. In the periphery, CPZ-feeding induced a dose-dependent suppression of splenic CD4/8 and organ mass, suggesting that this peripheral action of CPZ was a major impediment to studying the role of the peripheral immune system following demyelination and disruption of the BBB. Notably, oligodendrocytosis, demyelination and gliosis with the low dose of CPZ for 5 weeks resulted in minimal splenic atrophy and less severe adaptive immune system suppression, indicating that this might be a better model to test the ‘inside-out’ theory of MS. Moreover, using a highly sensitive ‘top-down’ proteomic approach, changes in 33 brain proteoforms were identified in the CPZ-fed mice, the majority of which were found to be associated with mitochondrial function. This strongly suggests that mitochondrial perturbations may elicit oligodendrocytosis and demyelination.

## Figures and Tables

**Figure 1 cells-08-01314-f001:**
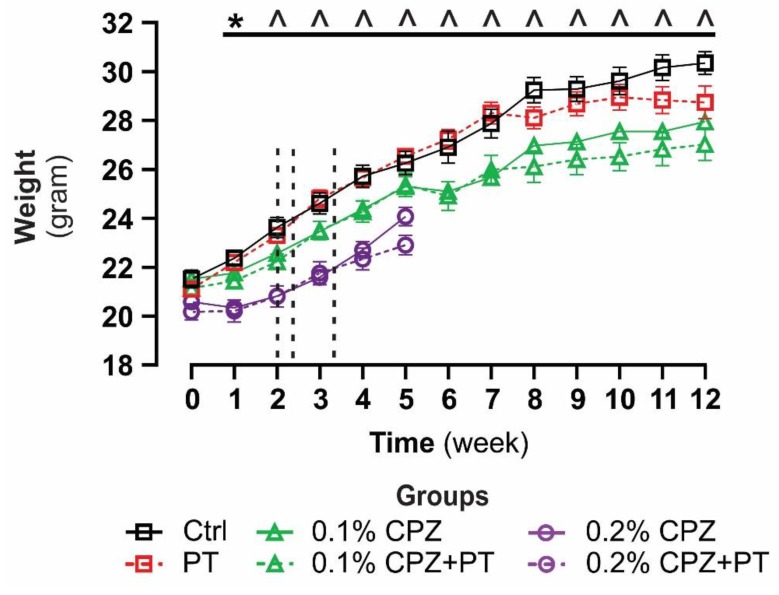
Body weight changes induced by CPZ-feeding. All mice gained weight over time. Groups fed the highest dose of CPZ(±PT) in each experiment gained weight significantly more slowly compared to other groups. Vertical dash lines indicate the timing of individual PT injections (i.e., days 14, 16, and 23). Data are expressed as mean ± SEM. Two-way ANOVA and Tukey post hoc analysis were used to determine differences among groups (* *p* < 0.05, ^ *p* < 0.0001, 5-week study *n* = 22 animals/group of which *n* = 12 animals/group continued feeding for 12 weeks).

**Figure 2 cells-08-01314-f002:**
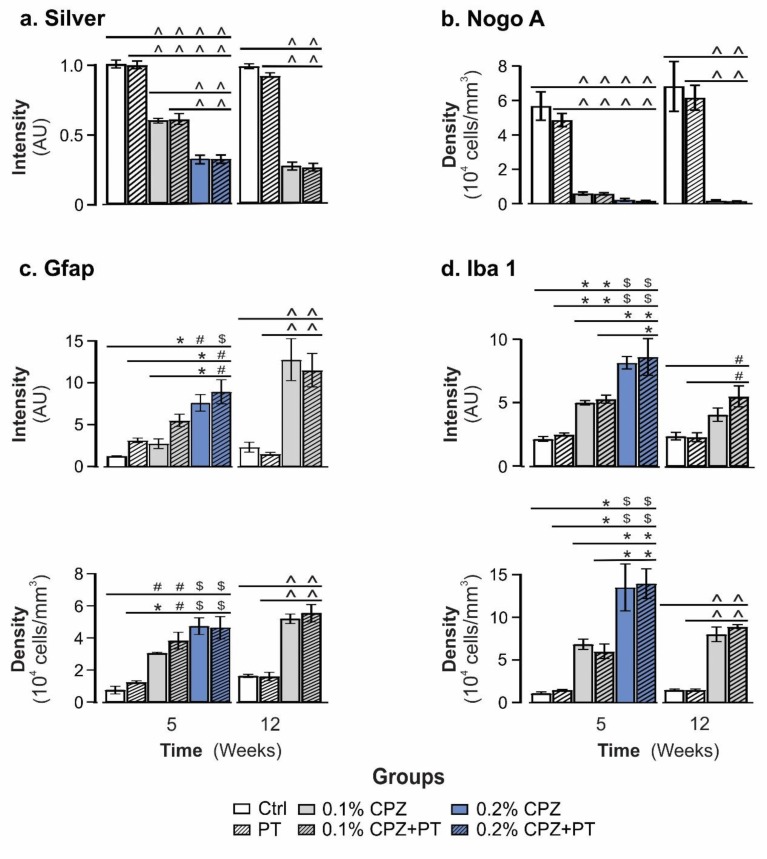
Quantification of demyelination, cell death and gliosis in the midline corpus callosum. (**a**) Silver staining. CPZ-feeding(±PT) led to significant demyelination and reduced silver staining intensity at 5 and 12 weeks. PT alone had no effect. Feeding 0.1% CPZ for longer (12 weeks) produced a comparable demyelination to that seen with 0.2% for 5 weeks. (**b**) Nogo A. CPZ-feeding(±PT) led to significant oligodendrocytosis with 0.1% CPZ was as effective as 0.2% CPZ at either time point. PT alone had no effect. (**c**) Gfap. Staining intensity increased in a dose dependent manner, in the CPZ(±PT) treated groups and this was associated with an increase the number of Gfap positive astrocytes at both time points. PT only did not evoke a Gfap response. (**d**) Iba 1. Increased Iba 1 fluorescence intensity and number of Iba 1 positive microglia were seen in both 5- and 12-week groups. PT alone produced no microglial response. Data are presented as mean ± SEM. One-way ANOVA and Tukey post hoc analysis was used to determine differences among groups (* *p* < 0.05, # *p* < 0.01, $ *p* < 0.001 and ^ *p* < 0.0001). Quantitation based on analysis of 5–9 sections/animal, 3–5 animals/group.

**Figure 3 cells-08-01314-f003:**
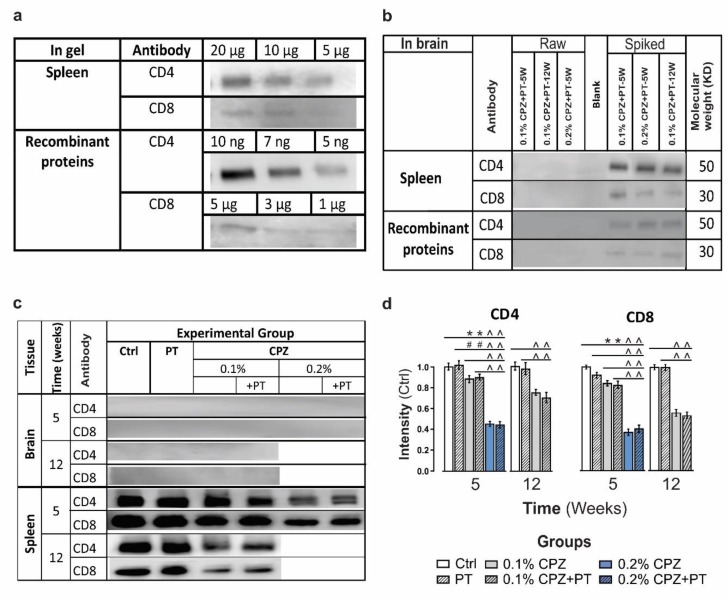
Western blot analysis. (**a**) Measurement of the detection limit of CD4/8 signal using naïve Ctrl spleen homogenates and CD4/8 recombinant proteins in gels. (**b**) Confirmation of CD4/8 signal detection using brain tissue samples (60 μg) spiked with either 5 µg or 10 µg of spleen homogenate or 5 ng and 5 µg commercial CD4 and CD8 recombinant proteins. (**c**) No CD4/8 signal was detected in CPZ(±PT) brain samples whereas a reduced CD4/8 signal intensity was found in spleen. (**d**) Quantification of splenic CD4/8 blots showed a significant reduction of CD4 and CD8 signal intensity in spleens of specific groups. Cropped CD4/8 western blots are presented unaltered and shown in their entirety in Appendix A. Data are presented as mean (±SEM) relative to the Ctrl mice. One-way ANOVA and Newman-Keuls Multiple Comparison post hoc analysis were used to determine differences among groups (* *p* < 0.05, # *p* < 0.01 and ^ *p* < 0.0001).

**Figure 4 cells-08-01314-f004:**
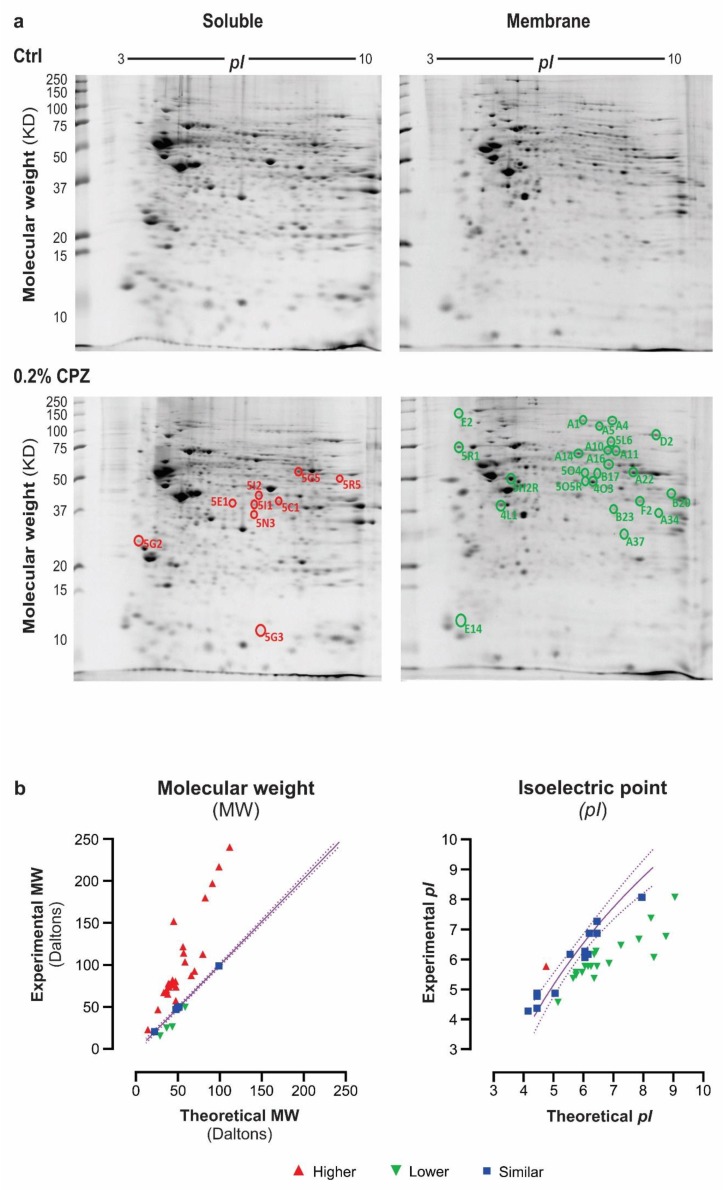
Top-down proteomic analysis. (**a**) Representative two-dimensional gel images of soluble (SP) and membrane (MP) brain proteomes from naïve Ctrl and 0.2% CPZ-fed groups used to detect proteoform changes. Proteoforms were resolved on the basis of their isoelectric point (*pI*) and molecular weight (MW). The total number of spots across different groups at both 5 and 12 weeks is given in Appendix A. Delta 2D software analysis revealed 33 unique spots for which the spot volume changed by at least 1.5-fold in at least one experimental group; soluble proteoforms are indicated by red circles and membrane proteoforms by green circles. The identities of the protein species are shown in Table 1 (*n* = 15 gels/fraction, n = 5 animals/fraction, 5-week study *n* = 180 gels and 12-week study *n* = 120 gels). (**b**) Comparison between theoretical and experimental MW (left) and *pI* (right) of identified proteoforms.  Red represents increase, green decrease and  blue indicates no statistically significant difference between the experimental and theoretical values. Purple dashed lines indicate 95% confidence intervals and the solid line represents full agreement between experimental and theoretical values.

**Figure 5 cells-08-01314-f005:**
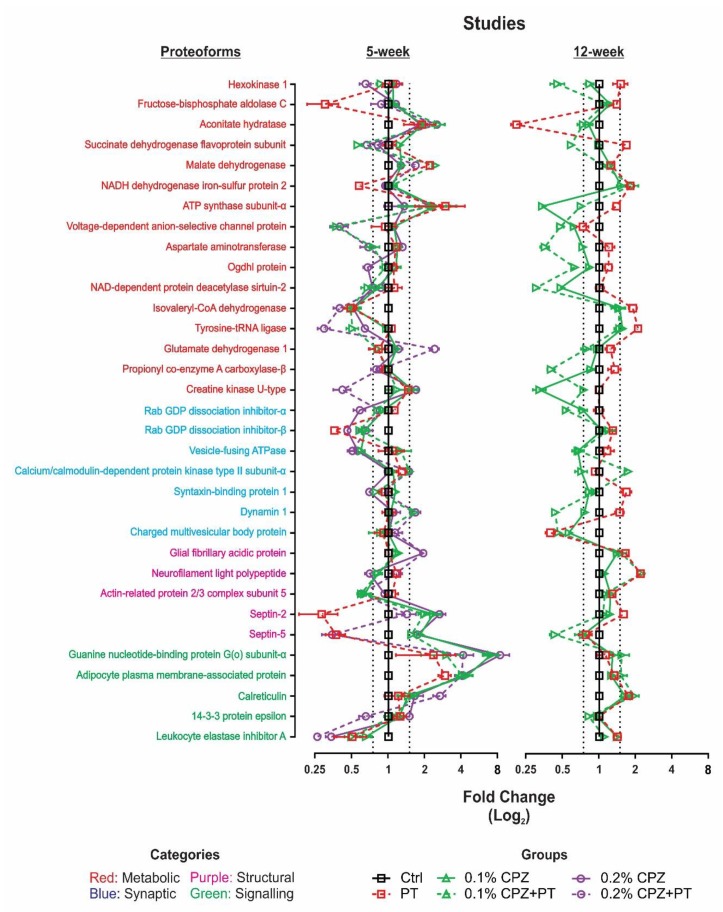
Log_2_ fold changes in abundance for 33 proteoforms relative to Ctrl. Each point is the average change in spot volume ratio from all treated groups (PT, 0.1% CPZ, 0.1% CPZ+PT, 0.2% CPZ and 0.2% CPZ+PT) relative to Ctrls from the triplicate gels resolved for each of the 5- and 12-week samples. Only significant changes in abundance (*p* < 0.05) that exceeded the established 1.5-fold criteria (dash lines) in at least 1 experimental group were selected for excision, processing, and protein identification. Solid and dashed lines connect the protein changes within each experimental group with and without PT injection, respectively. Proteoforms (top left) indicate the different functional categories including metabolic (red), synaptic (blue), structural (purple) and signalling (green). Analysis was based on n = 180 gels (5-week study) and, n = 120 gels (12-week study); n = 5 animals/group.

**Figure 6 cells-08-01314-f006:**
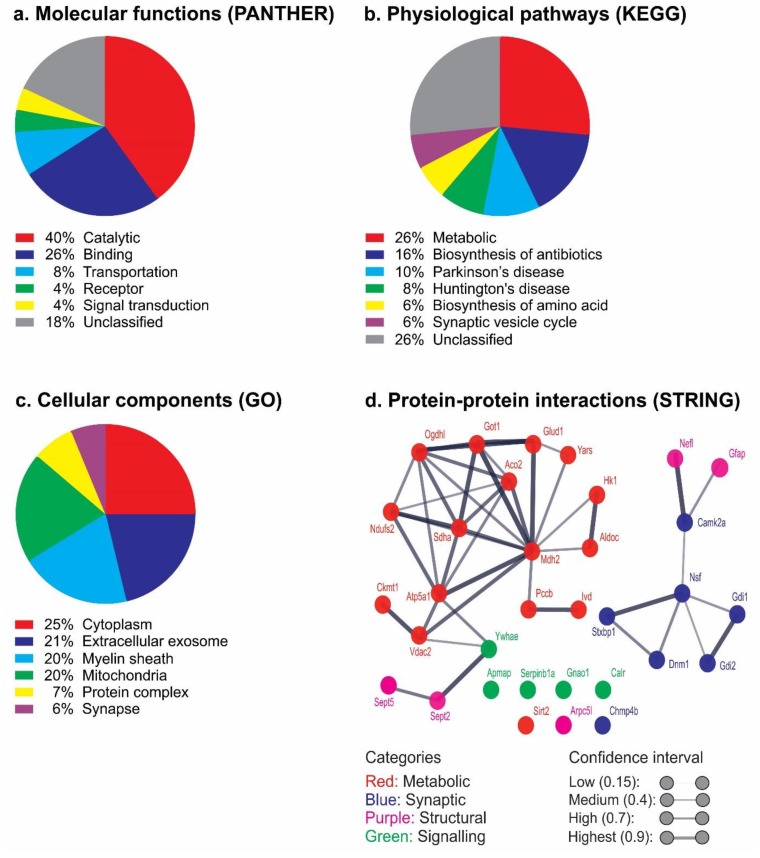
Functional clustering and protein–protein interactions. Pie charts show the distribution of proteins according to (**a**) Molecular functions (characterized using PANTHER), (**b**) Physiological pathways (categorised using KEGG), (**c**) GO cellular components. (**d**) Protein–protein interaction association network maps. The strength of connections is based on co-expression, gene fusion, co-occurrence, neighbourhood, databases, experiments, and text-mining collated in the STRING database. The strength of interactions is indicated by the thickness of the lines. STRING analysis revealed 4 protein clusters involved in metabolic (red), synaptic (blue), structural (purple) and signalling (green). Collectively, these proteins and their associations suggest that CPZ induced metabolic dysregulations and mitochondrial dysfunction.

**Figure 7 cells-08-01314-f007:**
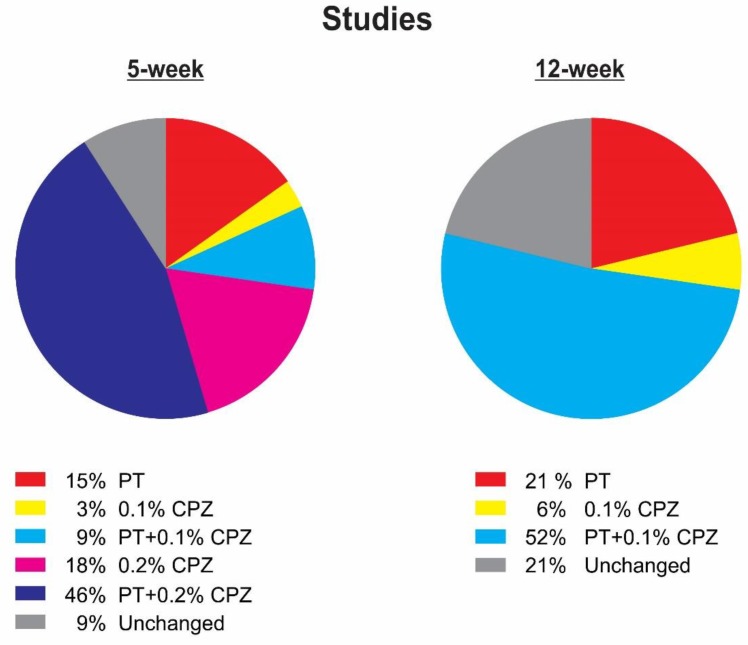
Highest fold change. Pie charts showing the highest increase or decrease fold change in abundance of the 33 proteoforms relative to Ctrls in the 5- and 12-week studies. PT alone or when PT was combined with CPZ showed greater change than CPZ alone. Quantification is based on ≥1.5-fold changes.

**Table 1 cells-08-01314-t001:** 2DE LC/MS/MS analyses identified 33 proteins from 5- and 12-week studies.

Spot ID/Tissue Fraction	UniProt ID	Protein Name	Gene ID	Score/Coverage %	Unique Peptides	MW/*pI*		Highest Fold Change		Data Base	Reference		
						Theoretical	Experimental	5 W	12 W		CPZ	EAE	MS
A4/MP	G3UVV4	Hexokinase 1	Hk1	270/10	10	101.8/6.2	220/6.4	0.66(0.2)↓	0.48(0.1PT)↓	M	-	-	[87]
5N3/SP	P05063	Fructose-bisphosphate aldolase C	Aldoc	510/33	12	39.3/6.6	73.4/5.9	0.32(PT)↓	×	S	-	[88]	[89]
D2/MP	Q99KI0	Aconitate hydratase	Aco2	407/25	18	85.4/8.1	182.9/8.2	2.37(0.2PT)↑	0.23(PT)↓	M	-	-	[90]
A10/MP	Q8K2B3	Succinate dehydrogenase flavoprotein subunit	Sdha	230/12	8	72.5/7	96/6	0.57(0.1PT)↓	0.61(0.1PT)↓	M	[91]	-	-
F2/MP	P08249	Malate dehydrogenase	Mdh2	983/57	25	36/8.9	70.2/6.9	2.30(0.1PT)↑	×	S	-	[88]	[92]
5O5R/MP	Q91WD5	NADH dehydrogenase iron-sulfur protein 2	Ndufs2	173/10	5	52.5/6.5	52.2/5.5	0.59(PT)↓	1.77(0.1PT)↑	M	[91]	-	-
5R5/SP	Q03265	ATP synthase subunit-α	Atp5a1	177/26	15	59.7/9.2	117.4/8.2	2.76(PT)↑	0.37(0.1)↓	S	[91]	-	[89]
A37/MP	Q60930	Voltage-dependent anion-selective channel protein	Vdac2	155/14	5	31.6/7.4	18.7/6.6	0.42(0.2PT)↓	0.5(0.1PT)↓	M	-	-	[90]
A34/MP	P05201	Aspartate aminotransferase	Got1	350/26	12	46.2/6.6	29.5/7.4	0.69(0.2PT)↓	0.38(0.1PT)↓	M	-	-	[89]
A1/MP	B2RXT3	Ogdhl protein	Ogdhl	291/10	10	114.5/6.4	243.3/5.9	0.69(0.2)↓	0.65(0.1PT)↓	M	-	-	-
B23/MP	Q8VDQ8	NAD-dependent protein deacetylase sirtuin-2	Sirt2	529/36	13	39.4/6.35	27.8/7	0.69(0.1PT)↓	0.33(0.1PT)↓	M	-	[93]	[93]
5C1/SP	Q9JHI5	Isovaleryl-CoA dehydrogenase	Ivd	650/39	17	46.3/8.5	79.4/6.2	0.42(0.2PT)↓	1.80(PT)↑	S	[49]	-	-
5G5/SP	Q91WQ3	Tyrosine-tRNA ligase	Yars	333/28	17	59/6.57	124.9/6.4	0.32(0.2PT)↓	1.97(PT)↑	S	-	-	-
A22/MP	P26443	Glutamate dehydrogenase 1	Glud1	324/12	7	61.3/8	106.8/6.8	2.24(0.2PT)↑	×	M	-	[94]	[95]
A16/MP	Q99MN9	Propionyl co-enzyme A carboxylase-β	Pccb	277/16	6	50.4/5.7	60.5/6.3	×	0.43(0.1PT)↓	M	-	-	-
B20/MP	P30275	Creatine kinase U-type	Ckmt1	137/22	11	46.9/8.4	83.2/7.5	0.44(0.2PT)↓	0.35(0.1)↓	S	-	-	-
A14/MP	P50396	Rab GDP dissociation inhibitor-α	Gdi1	227/20	8	50.5/4.9	77/5.9	0.6(0.2)↓	0.56(0.1PT)↓	M	-	-	[96]
5O4/MP	Q61598	Rab GDP dissociation inhibitor-β	Gdi2	148/10	5	50.5/5.9	55.6/5.7	0.38(PT)↓	×	M	-	-	[96]
5L6/MP	P46460	Vesicle-fusing ATPase	Nsf	466/36	33	82.6/6.5	116/6.3	0.52(0.2PT)↓	0.71(0.1PT)↓	S	-	-	-
B17/MP	P11798	Calcium/calmodulin-dependent protein kinase type II subunit-α	Camk2a	225/16	7	54/6.6	53.2/7	×	0.42(PT)↓	M	[97]	[98]	-
A11/MP	O08599	Syntaxin-binding protein 1	Stxbp1	244/12	5	68.7/6.3	90.6/6.3	0.71(0.2)↓	1.61(PT)↑	M	-	[94]	-
A5/MP	A0A0J9YUE9	Dynamin 1	Dnm1	172/9	8	93.9/6.2	200/6.2	1.61(0.2PT)↑	0.46(0.1PT)↓	M	[49]	-	-
E14/MP	Q9D8B3	Charged multivesicular body protein	Chmp4b	176/15	4	24.9/4.6	23.7/5	×	1.65(0.1PT)↑	M	-	-	-
5H2R/MP	P03995	Glial fibrillary acidic protein	Gfap	1802/72	45	49.8/5.2	83.7/5.0	1.86(0.2PT)↑	1.58(PT)↑	S	[99]	[100]	[101]
E2/MP	P08551	Neurofilament light polypeptide	Nefl	1594/57	92	61.4/4.6	53/4.9	0.71(0.2)↓	2.08(0.1PT)↑	S	-	[102]	[103]
5G3/SP	Q9D898	Actin-related protein 2/3 complex subunit 5	Arpc5l	152/26	4	16.9/6.3	26/5.9	0.62(0.2PT)↓	×	M	-	-	-
5I1/SP	P42208	Septin-2	Sept2	180/21	7	41.5/6.1	81/5.7	0.30(PT)↓	1.53(PT)↑	S	-	[93]	[104]
5I2/SP	Q9Z2Q6	Septin-5	Sept5	159/22	9	42.7/6.2	80.4/5.9	0.36(0.2PT)↓	0.46(0.1PT)↓	S	-	[100]	-
4L1/MP	P18872	Guanine nucleotide-binding protein G(o) subunit-α	Gnao1	186/15	5	40.6/5.3	68.5/4.7	7.35(0.2)↑	1.48(0.1PT)↑	S	-	[105]	-
4O3/MP	Q9D7N9	Adipocyte plasma membrane-associated protein	Apmap	148/8	5	46.4/5.9	85/5.6	4.09(0.1)↑	1.46(0.1PT)↑	M	-	-	-
5R1/MP	P14211	Calreticulin	Calr	187/13	7	47.9/4.3	155/4.4	2.50(0.2PT)↑	1.83(0.1PT)↑	S	-	[105]	[106]
5G2/SP	P62259	14-3-3 protein epsilon	Ywhae	110/23	6	29.1/4.6	49.9/4.5	0.66(0.2PT)↓	×	S	-	[100]	[107]
5E1/SP	Q9D154	Leukocyte elastase inhibitor A	Serpinb1a	191/36	18	42.5/5.8	76.7/5.5	0.28(0.2PT)↓	×	S	[49]	-	-

**Key**: MP, membrane protein; SP, soluble protein; MW, molecular weight; *pI*, isoelectric point; S, Swiss-Prot; M, MSPnr100; PT, pertussis toxin; 0.1, 0.1% CPZ; 0.1PT, 0.1% CPZ+PT; 0.2, 0.2% CPZ; 0.2PT, 0.2% CPZ+PT; W, week; ×, unchanged; ↑, increase; ↓, decrease; -, not found or investigated (details are shown in Figure 5). Some of the spots contained more than one clearly identifiable protein; presented here are the hits with the highest score, coverage and peptide count. UniProt and gene IDs were derived from the UniProt database. MASCOT score, sequence coverage, theoretical (MW/*pI*), and unique peptides number were acquired from the MASCOT database search. Experimental (MW/*pI*) was derived from 2D gels of identified protein spot. References are from the published literature in PubMed on CPZ, EAE and MS and used to compare currently identified proteins with the existing literature.

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
