# Peer review of "Suppression of the Peripheral Immune System Limits the Central Immune Response Following Cuprizone-Feeding: Relevance to Modelling Multiple Sclerosis"

_cells, 2019, doi:10.3390/cells8111314_

Round 1

Reviewer 1 Report

The authors used the cuprizone (CPZ) model to investigate the inside-out pathogenesis theory of MS, i.e. if chronic demyelination combined with disruption of the blood-brain barrier induces T cell responses. They also investigated the brain proteome by using two-dimensional gel electrophoresis coupled with LC-MS. They concluded that CPZ suppresses the peripheral immune responses therefore CD4 and CD8 cells do not enter the brain, and this is a major impediment to use this model for the inside-out theory. They identified tissue proteome changes and dysregulation of 33 proteins.

The data are interesting, but there are a few issues to be addressed.

This theory was investigated in a genetically modified mouse model of oligodendrocyte death (this work is cited, Traka, Nat Neurosci 2015). That article showed that almost a year is necessary to observe inflammatory responses. Did the authors continue observation after the 12-week CPZ treatment for a long period, i.e. for months? In the Introduction, the authors discuss the effect of CPZ on the atrophy of the spleen, and on the peripheral immune organs. They should mention here also the CPZ effect on the thymus (atrophy of a central immune organ) and the possible effect of T cell selection/development on the absence of adaptive immune responses in the brain. This work is cited and discussed in the Discussion, but logically it should be mentioned also in the Introduction (Solti, Plos One 2015). In figure 1, the 0.1% CPRZ treatment had no effect on the body weight up to 5 weeks (left panel), but on the right panel there is difference already at 1 week. Is this a mistake? Did they use any control to be sure that the blood-brain barrier was disrupted? Are there any evidences that the toxic compound CPZ modifies the functional properties of T cells? Can toxic CPZ induce e.g. anergy of T cells, or modify their migration capacity? How the 33 dysregulated proteins were identified is not clear. How did they calculate significant changes? Was there any targeted quantification and validation of the identified protein changes? Is there any possibility to identify possible cellular location of the protein changes (oligodendrocyte, astrocyte, microglia)?

Reviewer 2 Report

The authors sought to test the ‘inside-out’ vs. the ‘outside-in’ hypotheses for MS using a cuprizone (CPZ) mouse model +/- pertussis toxin (PT), which disrupts the blood brain barrier. To examine this hypothesis they characterized proteome changes in mouse models (and controls) exposed to cuprizone and/or pertussis toxin in various dosage levels and time points.  They applied 2D gel electrophoresis followed by digestion, LC-MS/MS, and database searching with MASCOT to identify proteins that are differentially regulated between experimental conditions.  They argue that low doses of CPZ at 5 weeks may be a more promising model to study the ‘inside-out’ theory of multiple sclerosis.  There data suggest that CPZ-feeding(±PT)  induced changes in the brain proteome related to suppression of  immune function, cellular metabolism, synaptic function and cellular structure/organization, and that demyelinating conditions in MS can be initiated in the absence  of adaptive immune system involvement.  Generally, the manuscript well written and the work thoroughly discussed and (despite some exceptions noted below) the methods section was very comprehensive and should enable reproduction of the experiments. However, based on the limited scope of the study (small sample size) some of the conclusions drawn may be overstated.  For example, they argue in the conclusions that endoplasmic reticulum perturbations may elicit oligodendrocytosis and demyelination, a conclusion based upon the characterization of a single ER stress chaperone. Plus, regarding the analytical methodology it seems surprising that despite observing ~1600-1700 proteins with 2D gel electrophoresis that only ~33 were found to be differentially regulated.  Considering the “sensitivity” of the analytical approach applied one would have thought that the proteome results would correlate with the dose/time-dependent occurrence of oligodendrocytosis, demyelination and gliosis observed.  Plus, they note the methods are adept to characterizing things like post-translational modifications, yet this information is not presented in the tables/figures.   It is best that the authors provide a thoughtful discussion regarding these topics and the limitations of the proteomics approaches applied. 

General criticisms.

1)      Overall, the introduction was very comprehensive, however, the addition of a few details would be helpful to the reader. Although the CPZ mouse model is widely used in MS research, it would be informative to briefly state within the introduction what type of molecule CPZ is and what is known about its mechanism of action. Regarding the latter, the information stated in the abstract (CPZ affects oligodendrocytes resulting in demyelination) and discussion (CPZ induces iron dyshomeostasis) could be presented in the introduction.

2)      The type of model system (i.e., mouse) should also be stated within the abstract, introduction, and conclusions.

3)      There is an excessive amount of self-citations.

4)      Matix Science appears to define top-down database mining with Mascot differently than that applied here because this search mode employs a ‘non cutting’ enzyme search. http://www.matrixscience.com/help/top_down_help.html.  The methods should simply be called “proteomics” or “2DGE” or else there terminology that is aligned with the search engine employed which here appears to be either “Bottom-up” or for proteoform mining “middle down”  http://www.matrixscience.com/nl/201906/newsletter.html

5)      Lines 301-303 and table 1. What is the rational for using two databases?  Table 1 reports the database from which the protein match was made but are these databases mutually exclusive?  For example, what happens when the data from a protein matching in SwissProt is searched against MSPnr100: is there a different result and how do the scores compare?  The way table 1 presents the data makes it look like a database was picked arbitrarily and without any meaningful rationale.

6)      Lines 310-311. “The highest MASCOT score (>100) with sequence coverage >5%; and 2) >4 unique matched peptides.” As well as any other points in which the MASCOT score is referenced.  A) What settings were used for MASCOT searches? This should be made explicit in the paper as the score itself can change with changes in parameters.  B) A primary feature of MASCOT reports is a significance value and score distributions.  These give confidence intervals (5%,1%,etc) for likelihood that a particular protein matches the data.  None of this information, which would signify how likely the assignment was made not due to random chance.  The MASCOT/MOWSE scores themselves are often accompanied by a score distributions diagram which compares all possible protein matches within a database.  Therefore, a high score by itself is not sufficient for validating a protein as many other proteins within the database can fall within a similar score range given the input data. 

7)      Table 1: In general the data analysis was very thorough, with the exception of proteoform characterization. The author claims that differences in theoretical and experimental molecular weights can be due to dimerization and/or additional PTMs such as phosphorylation or glycosylation.  Are any of these PTMs observed in the MS/MS data or are these inferences from literature?   On the other hand, the theory of dimerization state by the authors is not supported by the observed experimental mass.  Given that no intact protein data measurements were shown at the mass spectrometry level, it is hard to rationalize that a dimer/trimer/etc has formed.

8)      Table 1: Differences in pI. Can the authors explain why there are at times such a dramatic pI shift in some of the proteins observed?  For example, Isovaleryl-CoA dehydrogenase was found to be >100 times more acidic in the experimental result (pI = 6.2) vs the theoretical value (8.5).   Numerous examples of this inconsistency exist (Sdha, Mdh2, Ndufs2,Atp5a1,Glud1,etc.).

9)      The inclusion of a statistical comparison of the CPZ + PT groups to the PT only group is necessary.  Currently the respective treatment groups (CPZ with or without PT) are only compared to the control group (no treatment). The former comparison would help to tease out the individual contributions of CPZ (immune challenge/demylination) and PT (blood brain barrier disruption). This comparison is discussed at line 587 in the context of Figures 5 & 7, but not earlier, and the statistical comparisons are not made.

10)  Figure 1 title (Line 391) states that “behavioral changes” are induced by CPZ, however no behavioral data is presented.

11)  Figure 2: Line 402 states that “no difference in the extent of demylination were found between the MCC and LCC regions in any of the groups at either feeding duration” however, it is not clear which region (or if both) are represented in Figure 2.

12)  Line 411: The reference to “Figure 2a” appears to be mislabeled and should be Figure “2b”. Furthermore, Figure 2b is discussed in the text after panels 2c & 2d.

13)  The paragraph spanning lines 690-757 is very long and the points discussed would be better understood if it was broken into multiple paragraphs.

Following are minor grammatical errors:

·        CC is not defined at first use (line 94)

·        Line 121: insert “as” previously described…

·        LCC is not defined at first use (line 168)

·        Methods section: “Merck-Millipore” is misspelled throughout

·        Starting section 2.3.1 and following: Inconsistent spelling of “homogenization/homogenisation”

·        Line 294: 2.3 “KV” should be “kV”

·        Line 709-10: “increase abundance” (grammar)

·        Line 711: “in CPZ” à  “in the CPZ model”

Round 2

Reviewer 2 Report

Overall I think the scientific presentation is fine and the changes made regarding experimental details were appreciated. 

Regarding the issues I raised concerning vernacular used throughout the document ("top-down proteomics" and "proteoform").  I had hoped the authors would have adopted the term 2DGE which more accurately describes the approach used. However, it is clear the authors are dug in on this topic. 

I believe that the vast majority of proteomics field would accept this manuscript as a "bottom-up" method because it is employing the classical steps associated with it (i.e., enzyme digestion and database searching with MASCOT for protein inference from the identified peptides). However, I am not certain that it is necessary to demand that the authors conform to standards for what is truly just scientific vernacular/jargon.   

Author Response

We appreciate that the Reviewer recognizes that this is an issue of jargon, as there are no formally accepted or defined terms in the literature nor any centralized body to define and/or formalize terminology. As such, and with the definitive MDPI journal on this subject -Proteomes (https://www.mdpi.com/journal/proteomes%20) -  already having asked the senior author to twice serve as Guest Editor on Special Issues on Top-down Proteomics in 2017 (https://www.mdpi.com/journal/proteomes/special_issues/top_down_proteomics; editors Prof. Jens R. Coorssen and Dr. Alfred L. Yergey; noting Dr. Yergey as an internationally recognized mass spectrometrist who would know perhaps better than anyone the difference between top-down vs bottom-up) and 2019 (https://www.mdpi.com/journal/proteomes/special_issues/top_down_proteomics_in memory of Alfred Yergey; editors Prof. Jens R. Coorssen and Dr. Matthew P. Padula), we are quite confident not only in the terminology employed but also in the fact that the 'vast majority of the proteomics field' that understand analytical chemistry would agree that this is a genuinely Top-down approach. Perhaps it is the reviewer who is ‘dug in’  -  and we consider that a very unnecessary personal commentary.